# Mixed-precision weights network for field-programmable gate array

**Ninnart Fuengfusin** [1]*, **Hakaru Tamukoh**[1,2]

**1** Graduate School of Life Science and Systems Engineering, Kyushu Institute of Technology, Kitakyushu, Fukuoka, Japan, **2** Research Center for Neuromorphic AI Hardware, Kyushu Institute of Technology, Kitakyushu, Fukuoka, Japan

* fuengfusin.ninnart553@mail.kyutech.jp

## Abstract

In this study, we introduced a mixed-precision weights network (MPWN), which is a quantization neural network that jointly utilizes three different weight spaces: binary {−1,1}, ternary {−1,0,1}, and 32-bit floating-point. We further developed the MPWN from both software and hardware aspects. From the software aspect, we evaluated the MPWN on the Fashion-MNIST and CIFAR10 datasets. We systematized the accuracy sparsity bit score, which is a linear combination of accuracy, sparsity, and number of bits. This score allows Bayesian optimization to be used efficiently to search for MPWN weight space combinations. From the hardware aspect, we proposed XOR signed-bits to explore floating-point and binary weight spaces in the MPWN. XOR signed-bits is an efficient implementation equivalent to multiplication of floating-point and binary weight spaces. Using the concept from XOR signed bits, we also provide a ternary bitwise operation that is an efficient implementation equivalent to the multiplication of floating-point and ternary weight space. To demonstrate the compatibility of the MPWN with hardware implementation, we synthesized and implemented the MPWN in a field-programmable gate array using high-level synthesis. Our proposed MPWN implementation utilized up to 1.68-4.89 times less hardware resources depending on the type of resources than a conventional 32-bit floating-point model. In addition, our implementation reduced the latency up to 31.55 times compared to 32-bit floating-point model without optimizations.

## Introduction

A convolutional neural network (CNN) has attracted attention owing to its abilities to achieve the state-of-the-art results in image recognition [1], semantic segmentation [2], and object detection [3]. One of advantage of a CNN is its scalability which allows it to increase its parameters to operate with larger and more complex data. For example, LeNet-5 [4], one of first CNN models, was proposed with 60,000 learnable parameters to operate on a handwritten digit dataset, MNIST [5]. The AlexNet [1] model, the winner of the ImageNet Large Scale Visual Recognition Challenge (ILSVRC) 2012 [6], increased the number of parameters to 62 million. This number has further increased to 144 million with VGG-19 [7], which was proposed for ILSVRC 2014.

**Data Availability Statement:** Programming code is available at https://github.com/ninfueng/mpwn.

**Funding:** This work was supported by JSPS KAKENHI Grant Number 17K20010, URL https://kaken.nii.ac.jp/en/grant/KAKENHI-PROJECT-17K20010/. The funders had no role in study

design, data collection and analysis, decision to publish, or preparation of the manuscript.

**Competing interests:** The authors have declared that no competing interests exist.

With an increasing trend in the number of parameters of a CNN, deploying a large-scale CNN in embedded devices has become challenging due to the memory and hardware limitations of edge devices. To solve these problems, various approaches have been proposed, including network pruning [8], a quantization neural network (QNN) [9–11], knowledge distillation [12], and efficient architecture design [13, 14].

In this paper, we focus on the QNN approach. In general, a QNN reduces the bit width of CNN parameters to less than a conventional 32-bit. The immediate effect is to reduce the overall memory footprint of the model. Another effect is that because the conventional data type of a CNN is floating-point, which is not suitable for implementation in hardware due to its complexity, a QNN can restrict its parameters to a hardware-friendly data type, such as the fixed-point or integer format [15].

The concept of the QNN is possible due to the ability of neural network to dynamically adapt its parameters during training to minimize the quantization loss. For example, one of QNN, BinaryConnect (BC) [9], reduces the bit width of the weight from 32-bit floating-point to 1-bit without significant loss in accuracy. However, in terms of computation, to utilize the advantages of precision reduction, specialized hardware is necessary to exploit the specific data types, as the commercial central processing unit (CPU) and graphics processing unit (GPU) do not support fixed-point arithmetic or data types.

Another problem in the QNN is that reducing the bit width has a trade-off with the quantization error. The higher the quantization error, the lower the performance of the QNN. To handle this trade-off, we propose a mixed-precision weights network (MPWN). The MPWN is designed to exhibit performance close to that of a conventional 32-bit floating-point model while maintaining a low bit width and other properties of QNN. The MPWN is a QNN that consists of three different weight spaces: binary $\{-1, 1\}$, ternary $\{-1, 0, 1\}$, and, 32-bit floating-point.

The MPWN assigns one of these weight spaces to the weight layer by considering on the order of layer or number of parameters within the layer. With three possible weight spaces per weight layer, the search range increases exponentially when the number of layers increases. Each search is expensive, as it requires time and resources to train a model from scratch. Therefore, finding an optimal combination by random search may not be effective. We address this problem with an accuracy sparsity bit (*ASB*) score, which quantifies the quality of the MPWN model in terms of three desired properties: accuracy, sparsity, and number of bit. Using a single scalar score enables the searching with Bayesian optimization (BO).

Another objective of this study is to demonstrate the effectiveness of the MPWN in hardware implementation. Therefore, we implemented MPWN in a field-programmable gate array (FPGA). One advantage of an FPGA over the conventional software language is it enables bitwise manipulation. Another advantage is that it allows the user to define, store, and compute an arbitrary data type. However, the main disadvantage of an FPGA design is a long development time. To gain advantages from both software and hardware, high-level synthesis (HLS) has been developed. HLS is a development platform that converts C-like languages (C, C++, and System C) to the hardware description language (HDL) (Verilog and VHDL) language. This allows users to rapidly develop applications with an interface of a C-like language with fewer constraints from the HDL language. With this property, HLS is especially useful when applies HLS with the deep learning algorithm that its state-of-the-art algorithm has been rapidly changed. A drawback of HLS compared to optimized handcrafted HDL is that HLS-generated HDL code may cause a higher latency and hardware utilization. For instance, Ordaz et al. [16] compared between HDL and HLS implementations of cyclic redundancy check (CRC) and found out that HDL implementation consumes less than LUT by 1.58 times and less amount of latency by 2.63 times.

In this study, we utilized Xilinx Vivado HLS (VHLS) [17] for hardware synthesis and Xilinx Vivado for hardware implementation. We demonstrate that by exploiting the weight spaces of the MPWN, we can reduce the hardware utilization of multiplication by replacing it with XOR signed bits (XSB) and ternary bitwise operation (TBO) [18]. XSB is a VHLS algorithm for XOR between signed bits of operands. TBO is also the same algorithm with XSB with an ability to detect zeros. If TBO detects input as zero, it will also output as zero. We demonstrate that XSB, TBO, and a specific data type can be used to significantly reduce the overall latency and hardware resources of the model compared with directly utilizing floating-point arithmetic only.

In prior works, there are at least two cases to deploy a mixed precision model. The first case is due to a limitation of low-precision models, for instance, Nakahara et al. [19] proposed a binarized YOLOv2 [20] designed for FPGA implementation. However, binarized neural network (BNN) [10] or a QNN with both binary weights and activations does not perform a bounding box prediction or regression task effectively. To address this problem, Nakahara et al. assigned the last layer of binarized YOLOv2 as the floating-point instead. Another case is to improve the performance of the low-precision model to be closer to a 32-bit floating-point model. For instance, Chu et al. [21] proposed a quantization method to progressively reduce the bit-width from the input to last layer. This method is realized from an observation that feature distributions in the shallow layer contain a low quantity of class separability while in the deeper layers, the distributions have a high quantity of class separability. Wang et al. [22] proposed a method using reinforcement learning to search suitable bit-widths in the layer-wise direction. By using with a hardware simulator, the energy and latency of the quantized model were utilized as direct feedbacks to the reinforcement learning controller. From these two categories, our work is categorized in the second category. The objective in this study is to achieve the performance of a 32-bit floating-point model while maintaining the properties of QNNs. To the best of our knowledge, comparing to previous researches in the mixed-precision network field, our novelty is we utilized a BO to search a suitable quantization layer instead of using the reinforcement learning or differentiable architecture search [23]. We provide *ASB* score that we specifically designed for the weight spaces. We included a sparsity as a part of *ASB* and we also left a choice to not quantization into the search space.

In terms of prior works in the FPGA direction, several works focus on QNNs with both weights and activations as either binary, fixed-point, and floating-point. For instance, FINN [24] and GUINNESS [25] are frameworks to construct BNN to the FPGA. Both FINN and GUINNESS utilizes HLS as a backend component to deploy BNN models into FPGA. Rongshi et al. [26] and Cho et al. [27] also utilized HLS to construct a floating-point and fixed-point CNN, respectively. Comparing with prior works in the FPGA field, to the best of our knowledge, our novelty is we provide a first FPGA implementation of binary or ternary weights model with floating-point activation. To effectively deploy with binary or ternary weights and floating-point activations, we also introduce XSB and TBO to replace floating-point multiplications with bitwise operations.

The main contributions of this study are as follows:

- We further evaluated the MPWN on the Fashion-MNIST [28] and CIFAR10 [29] datasets.

- We provided a more insightful analysis of MPWN weight spaces and heuristic rules with grid search for all possible combinations of the MPWN.

- We proposed the *ASB* score which makes it possible to systematically search for the optimal combination of the MPWN with BO.

- We designed XSB, a replacement for multiplication between floating-point and binary values 1, −1 for VHLS implementation.

- We synthesized and implemented the MPWN in an FPGA and demonstrated its effectiveness in terms of both latency and area.

- We published our programming codes on https://github.com/ninfueng/mpwn.

The remainder of this paper is structured as follows. Section 2 explains the building blocks of our study, including the QNNs and directives. Section 3 introduces the MPWN model, *ASB* score, and hardware design of the MPWN, and Section 4 presents the experimental results of our MPWN simulation, hardware synthesis, and hardware implementation. Section 5 concludes the paper.

## Quantization neural networks and FPGA

In this section, we describe two related QNNs that are applied as parts of the MPWN. In addition, we introduce fundamental VHLS concepts that are applied in later sections.

### BinaryConnect

BinaryConnect (BC) [9] is a QNN that binarizes its weights to the set {−1, 1}. With 2 possibilities, a binarized weight can be represented with a 1-bit. The BC quantization equation is expressed as Eq (1), where $i$ is the index of the weight layer, $W$ is the floating-point weight, and $W^b$ is the binarized weight:

$$W_i^b = \begin{cases} 1, & W_i \geq 0, \\ -1, & W_i < 0 \end{cases} \tag{1}$$

However, Eq (1) cannot be used for back-propagation. Eq (1) causes the gradient to be zeros everywhere except at 1 and −1. To modify this function to be trainable, BC overwrites the back propagation of Eq (1) to Eq (2), where $L$ is the loss function. Eq (2) allows gradients to be able to pass through Eq (1) in the same manner as an identity function:

$$\frac{\partial L}{\partial W_i} = \frac{\partial L}{\partial W_i^b} \tag{2}$$

In BC, there is an additional modification, or weight clipping, to the updating equation as illustrated in Eq (3), where $\eta$ is the learning rate. Eq (3) clips the updated weights into the range of [-1, 1]. The authors of [9] indicated that this pushes the weights back into the active region.

$$W_i = min\left( max\left( W_i - \eta \frac{\partial L}{\partial W_i}, -1 \right), 1 \right) \tag{3}$$

### Ternary weight network

A ternary weight network (TWN) [11] is a QNN that quantizes its weights in each layer to the set {−$S_i$, 0, $S_i$}, where $S_i \in \mathbb{R}$, $W^t$ represents the ternarized weights, and $i$ is the order of layer. $S_i$ can be determined using Eq (5). The TWN quantization equation is presented in Eq (4), where

$\Delta$ is a threshold that can be determined by using Eq (6).

$$W_i^t = \begin{cases} S_i, & W_i > \Delta_i, \\ 0, & W_i \le \Delta_i, \\ -S_i, & W_i < -\Delta_i \end{cases} \tag{4}$$

$$S_i = E_{i \in \{i \mid |W_i| > \Delta\}}(|W_i|) \tag{5}$$

$$\Delta_i = 0.7 \times E(|W_i|) \tag{6}$$

Compared to BC with an additional zero in the weight space or with higher precision, the TWN has higher performance or accuracy. However, this doubles the number of bits to represent its weight becomes twice. With an additional zero, TWN introduces the concept of sparsity. Sparsity is defined as the number of zeros in a given array divided by the number of all parameters in the array. Back-propagation of the TWN faces the same problem as back-propagation of BC. The TWN solves this problem with the same method as BC using the redefined back-propagation of the quantization equation as expressed in Eq (7):

$$\frac{\partial L}{\partial W_i} = \frac{\partial L}{\partial W_i^t} \tag{7}$$

## Directives and hardware design in Vivado high-level synthesis

There are several metrics for hardware synthesis and implementation, including latency, hardware utilization, and power consumption. In terms of hardware resources in recent FPGAs, four fundamental hardware resources are block RAM (BRAM-18K), DSP48F block, flip-flop (FF), and look-up table (LUT).

One advantage of an FPGA over a CPU is that it can be designed to perform large-scale parallelism. Although parallelism greatly reduces the latency, the trade-off is that it also significantly increases hardware resources. Since VHLS receives a C-likes language as input and C-like languages are not designed for hardware implementation, VHLS solves this problem by providing users with hints regarding how to convert C-type functions into hardware. VHLS provides directives or *pragma* in C-like languages as hints. Using the directives, we can communicate to VHLS how to manage parallelism, memory, and other aspects.

There are three directives that we utilize throughout this study. The first two directives are related to parallelism, while the third directive is related to memory management. The first directive is the *PIPELINE* directive. Placing *PIPELINE* directives after a *for loop* indicates to VHLS that the *for loop* can be accelerated by data pipeline parallelism. The second is the *UNROLL* directive, which indicates that the computation in the *for loop* can be executed in parallel across the *for loop*. The third directive, the *ARRAY_PARTITION* directive, is mostly applied with either *PIPELINE* or *UNROLL*. Performing parallel computing requires accessing multiple data at the same time. By using *ARRAY_PARTITION*, a large memory is divided into multiple smaller blocks of memory. Therefore, it can access smaller memory in the same clock cycle without access conflict. With these three mentioned directives, we can apply parallel computing throughout our model to accelerate computation and reduce latency.

## Mixed-precision weights network and FPGA design

In this section, we describe our proposed method, MPWN, in terms of both algorithm and hardware design. This includes the MPWN algorithm and its effective design in hardware.

### Mixed-precision weights network

Mixed-precision weights network (MPWN) is designed to utilize the advantages of the weight spaces from BC, TWN, and 32-bit floating-point. The MPWN is partly motivated by dropout [30], which is a regularization method that randomly drops weight connections of a neural network in each batch training. Randomly dropping weights reduces the tendency of neurons to cooperate with other neurons, which leads to over-fitting. The optimal dropout rate is not always equal throughout the weight layers. The optimal rate depends on the layer order and the type of weight layer (e.g., convolutional layer, fully-connected layer). The difference in the optimal dropout rate reveals the sensitivity of the weight layer to the constraint. Disturbing a high sensitivity weight layer causes a greater negative effect in terms of performance than disturbing a low-sensitivity layer. Using this concept, the MPWN places different constraints depending on the sensitivity of the weight layers.

For hardware implementation, the performance of the TWN is still acceptable without the scaling factor $S_i$. Therefore, we utilize the TWN without the scaling factor, which also reduces the overall complexity of the hardware implementation. During inference, the 32-bit floating-point model can be reduced to the 16-bit floating-point without decreasing in performance. Therefore, we utilize with 16-bit floating-point instead of 32-bit floating-point.

We define the notations for representing MPWN layers as follows: **F** indicates that layer is 16-bit floating-point or full-precision, **B** indicates that the layer is the BC, and **T** indicates that the layer is TWN without the scaling factor. Therefore, **FBT** refers to a CNN with three layers. The first layer is 16-bit floating-point, the second layer is BC, and the third layer is TWN. The advantage of each weight space is summarized in Table 1. Overall, **F** is correlated with the performance of the model. **B** reduces the bit width of model the most, and **T** introduces sparsity into the model.

With several metrics to optimize in the MPWN, we reduce these scores to a single score called the accuracy sparsity bit (*ASB*) score. The *ASB* score is a linear combination of accuracy, sparsity, the number of bits, as expressed in Eq (8), where *a* is the model test accuracy, *s* is the sparsity of the model, and $b_{nor}$ is 1 minus the normalized number of bits. $b_{nor}$ is defined in Eq (9), where $b_{max}$ is the number of bits from the 16-bit floating-point model and *b* is the number of bits of given MPWN. Using Eq (9) instead of using only $\frac{b}{b_{max}}$, the optimization direction reverses from minimization to maximization which is the same direction as *a* and *s*.

$$ASB = \frac{a + s + b_{nor}}{3} \tag{8}$$

$$b_{nor} = 1 - \frac{b}{b_{max}} \tag{9}$$

**Table 1. Overview of properties of each mixed-precision weights network weight space.**

|  | Accuracy | Sparsity | Bits per Weight (bit) | Weight space |
|---|---|---|---|---|
| Full precision weights **(F)** | **High** | None | 16 | $\mathbb{R}$ |
| Ternary weights **(T)** | Mid | **High** | 2 | $\{-1, 0, 1\}$ |
| Binary weight **(B)** | Low | None | **1** | $\{-1, 1\}$ |

In Eq (8), $a$, $s$, and $b_{nor}$ does not contribute equally due to different in distributions. One of the metrics in $ASB$ may distribute with a low variance relative to other metrics. Therefore, the metric with low variance contributes to $ASB$ less than others. One of the factors of this low variance issue is a narrow range between the minimum and maximum values of the metric. To solve this issue, we introduce a min-max normalization to re-distribute each metric of $ASB$ into the same range [0, 1]. The min-max normalization is defined in Eq (10) where $x$ is an original value, $x_{max}$ is a maximum value in the distribution, $x_{min}$ is a minimum value in the distribution, and $x_n$ is a normalized value.

$$x_n = \frac{x - x_{min}}{x_{max} - x_{min}} \tag{10}$$

Finding maximum and minimum values in the worst-case is required to train all possible combinations of MPWN which is not feasible with a large model. Therefore, we introduce a method to estimate the maximum and minimum values of $a$, $s$, and $b_{nor}$ instead. This method is called as estimating rules which are formulated from properties from Table 1 and observations from grid-search with all possible combinations of MPWN with the LeNet-5 model. From this grid-search, the **F** model contains the highest amount of $a$ and the lowest amount of $b_{nor}$ and $s$. The **B** model contains the highest amount of $b_{nor}$ and close to the lowest amount of $a$. The **T** model contains almost the highest amount of $s$. With these properties, we formulate the estimation rules are as follows:

- **F** model contains a maximum value of $a$ while **B** model contains a minimum value of $a$.

- **B** model contains a maximum value of $b_{nor}$ while **F** contains a minimum value of $b_{nor}$.

- **T** model contains a maximum value of $s$ while **F** contains a minimum value of $s$.

By using these estimation rules, we can estimate the maximum and minimum values for $a$, $s$, and $b_{nor}$ by only using information from **F**, **B**, and **T** models. Using the min-max normalization improves the variance issue to an extent by improving the range of the metric with low variance. To further improve on the variance situation, one of the methods to increase or decrease the variance of a distribution is to multiply with a constant. Therefore, we can balance the variance of each metric of $ASB$ by modifying Eq (8) to a weighted average as shown in Eq (11) using with weights: $\alpha$, $\beta$, and $\gamma$. We can convert Eq (11) back to Eq (8) by using $\alpha = 1, \beta = 1$, and $\gamma = 1$. $\alpha$, $\beta$, and $\gamma$ can be adjusted to indicate what degree the $a$, $s$, and $b$ contribute to $ASB$.

$$ASB = \frac{\alpha a + \beta s + \gamma b_{nor}}{\alpha + \beta + \gamma} \tag{11}$$

The goal of the MPWN is to maximize the vector of quantization layers **l**, as shown in Eq (12), where $\mathbf{l} \in l^n$, $l \in \{\mathbf{F}, \mathbf{B}, \mathbf{T}\}$, and $n$ denotes the number of weight layers.

$$l^* = \text{argmax}_l ASB \tag{12}$$

The computational complexity of identifying the global maximum of the MPWN is $O(3^n)$. Each iteration of the search is expensive in terms of training time. To avoid examining all possible combinations of layers, in a previous study [31], we proposed human-based knowledge rules or three heuristics rules to identify a reasonable optimized **l**. The heuristic rules are as follows:

1. Layers that contain a large number of weight parameters should be **T**.

2. Layers that contain a small number of weight parameters should be **B**.

3. The first and last layers should be **F**.

In this case, we define a large number as a number that more than one positive standard deviation from a mean and we define a small number as a number that less than or equal to one positive standard deviation from the mean. To provide an example, we apply these heuristic rules to LeNet-5 [32]. The result is the **FBTBF** model since the third layer of LeNet-5 is a fully-connected layer with the number of weights that exceeds one standard deviation from the mean. These heuristic rules originate from several observations. For the first rule, placing **T** into a layer with the largest number of weight parameters causes the high sparsity in the model. Therefore, by only fixing **T** to certain layers, we can optimize other layers with other types of weight spaces. The second rule states that the default weight layer should be **B** to optimize the number of bits. **B** is the most suitable in terms of hardware implementation. The third rule states that placing **F** into the first layer significantly improves the accuracy. The last layer affects the confidence of the prediction, therefore, we also select the last layer as **F**.

The three heuristic rules allow the MPWN to perform a single search to find a suitable combination for the *ASB* score; however, the heuristic rules do not guarantee a global maximum. Systematizing the search process using the *ASB* score allows BO to be used. BO is conventionally applied to search the optimal hyperparameters. BO is summarized in [33]. In general, BO optimizes a given cost function with two objectives. The first objective is to explore the function and map the surrogate model from the obtained information. The second objective is to search for a local optimal location of the given function. BO is suitable for our layer search from the two following aspects:

- BO does not require gradient; therefore we can optimize our MPWN model with the sparsity and number of bits, which are not differentiable metrics.

- BO is defined with the constraint that each iteration is expensive to evaluate which meets the requirements of our problem [33].

## 1- and 2-bit signed integer

To fully utilize an FPGA with the MPWN, VHLS provides support to access arbitrary data types, such as signed integer (*int*), unsigned integer (*uint*), and the fixed-point (*fixed*) data type. To optimize the data type of **B** and **T**, we utilize 1-bit (*int1*) and 2-bits signed integer (*int2*), respectively. However, the range of *int1* can contain only in the set of $\{-1, 0\}$, which does not cover 1 in the **B** weight space. To address this problem, we replace 1 with 0. This does not affect the performance of the MPWN implementation, as the replacement still holds the same signed bit information that is used in XSB. Reducing the number of bits with a specific data type promises a significant reduction in memory resources in our FPGA implementation.

## Half-precision floating-point

VHLS supports another data type: a half-precision floating-point or 16-bit floating-point (*half*). A QNN displays the robustness of a CNN against reducing precision, and we exploit this property by assigning the *half* data type to our MPWN model. Compared with 32-bit floating-point (*float*) and 64-bit floating-point (*double*) in terms of multiplication of two variables with of the same data type, *half* has the potential to reduce both the hardware resources and latency, as illustrated in Table 2. Table 2 presents the results of the implementation generated by VHLS, where the target device is Zynq UltraScale+ MPSoC ZCU102 or xczu9eg-ffvb1156-

**Table 2. Latency and hardware resources for multiplication of two variables with the same data type.**

| Data type | Latency (Clock cycle) | Hardware Resource | | |
|---|---|---|---|---|
| | | DSP48E | FF | LUT |
| *double* | 4 | 11 | 304 | 236 |
| *float* | 1 | 3 | 130 | 150 |
| *half* | **1** | **2** | **66** | **49** |

2-i. Table 2 demonstrates that *half* can reduce the resources to roughly half of those of *float* to one-fifth of those of *double*.

## XOR signed-bits

The MPWN may consist of multiple **B**. Therefore, it is necessary to further optimize the computation in the weight space. The multiplication between *float* and binary values {−1, 1} causes a sign to be changed. However, without a specific design, the multiplication between them is treated as a floating-point multiplication that consumes more resources than necessary. XOR signed bits (XSB) is designed as a replacement for multiplication between *float* and binary values. Multiplication between these variables is reduced with the XOR operation between the sign bits of the two operands, as illustrated in Fig 1.

In general, implementing XSB in HDL is simple whereas implementing it in VHLS is complicate. We present an XSB algorithm in C++ for VHLS, as illustrated in Listing 1. VHLS provides C++ libraries that support bitwise manipulation; however, the manipulation is constructed as methods within a built-in data type in VHLS. The problem that we face is that we cannot apply these methods directly with an unsupported data type (*double*, *float*, and *half*). Therefore, we are required to first convert an unsupported data type to bitwise supported data type. Then, we perform bitwise manipulation of the selected bit and convert it back to the unsupported data type.

**Listing 1**. **XOR signed-bits for Vivado high-level synthesis**.

```
void xor_signed_bits(int1 w, half x, half &out)
{
    #pragma HLS INLINE OFF
```

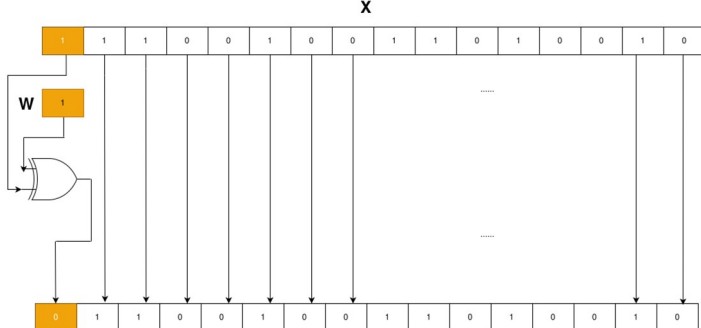

**Fig 1. XOR signed bits.** The top binary row presents a binary representation of the *half* data type, which represents a value of −123. The second binary row displays a binary representation of *int1*, which represents −1. By XOR only the most significant bit from both rows, the result is 123, which is the same as the answer to the general floating-point multiplication.

```
    int16 tmp_x;
    int16 tmp_out;
    // Convert half floating point to int16-
    // to access ap int built-in method.
    // Use uint1 because to cover {0, 1}.
    tmp_x = *reinterpret_cast<int16*>(&x);
    // XOR between signed-bit between weight and activation.
    uint1 sign = tmp_x.sign()^ w.sign();
    // Get bits from 14 to 0, not include the signed bit.
    int15 notsign = tmp_x.range(14, 0);
    // Concate between XOR result and concatenate-
    // between the rest of activation bits.
    tmp_out = sign.concat(notsign);
    out = *reinterpret_cast<half*>(&tmp_out);
}
```

Here, *fixed* is considered to apply instead of *half*. However, VHLS provides the *fixed* implementation with a binary representation as 2's complement, which is not compatible with the XSB algorithm. Toggling the sign of *fixed* requires reversing all the bits of *fixed* and subtracting by 1, which is an expensive process. Therefore, we apply our MPWN with *half* instead of *fixed*.

## Ternary bitwise operation

Ternary bitwise operation (TBO) was proposed by Honda et al. [18] as a replacement of multiplication with a ternary weight. TBO utilizes only an XOR and 15 AND gates as illustrated in Fig 2. TBO utilizes AND gates to detect whether the ternary weight is a zero or one (both positive and negative). If the weight is zero, the AND gates reset the variable to zero. Otherwise, it lets the variable pass through. Using the same concept of XSB, we can implement TBO in VHLS as shown in Listing 2 below.

**Listing 2. Ternary bit-wise operation for Vivado high-level synthesis**.

```
void xor_signed_bits (int2 w, half x, half &out)
{
    #pragma HLS INLINE OFF
    int16 tmp_x;
```

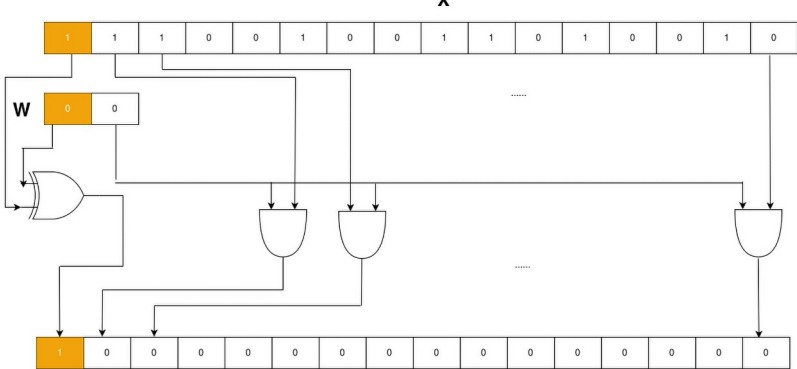

**Fig 2. Ternary bitwise operation.** The top binary row presents a binary representation of the *half* data type, which represents a value of −123. The second binary row displays a binary representation of *int2*, which represents 0. By using XOR and AND gates, the result is −0.

```
        int16 tmp_o;
        tmp_x = *reinterpret cast<int16*>(&x);
        // XOR between signed-bit between weight and activation.
        uint1 b15 = tmp_x.sign() ^w.sign();
        uint1 w0 = w.range(0, 0);
        // AND between OR results and rest of activation bit.
        uint1 b0 = w0 && tmp_x.range(0, 0);
        uint1 b1 = w0 && tmp_x.range(1, 1);
        uint1 b2 = w0 && tmp_x.range(2, 2);
        uint1 b3 = w0 && tmp_x.range(3, 3);
        uint1 b4 = w0 && tmp_x.range(4, 4);
        uint1 b5 = w0 && tmp_x.range(5, 5);
        uint1 b6 = w0 && tmp_x.range(6, 6);
        uint1 b7 = w0 && tmp_x.range(7, 7);
        uint1 b8 = w0 && tmp_x.range(8, 8);
        uint1 b9 = w0 && tmp_x.range(9, 9);
        uint1 b10 = w0 && tmp_x.range(10, 10);
        uint1 b11 = w0 && tmp_x.range(11, 11);
        uint1 b12 = w0 && tmp_x.range(12, 12);
        uint1 b13 = w0 && tmp_x.range(13, 13);
        uint1 b14 = w0 && tmp_x.range(14, 14);
        // Concatenate between all resultant bit.
        uint15 b_con = (b14, b13, b12, b11, b10, b9, b8, b7, b6, b5, b4, b3, b2, b1, b0);
    tmp_o = b15.concat(b_con);
    out = *reinterpret_cast<half*>(&tmp_o);
    }
```

## Overview of FPGA implementation

Using the *ASB* with *BO* allows us to find more optimized MPWN combinations than a combination from heuristic rules. Therefore, we implemented the model **FTTTF** that achieves the highest *ASB*. Our implementation of **T** was implemented with TBO instead of sparse matrix multiplication and convolutional operation due to the high overhead of sparse matrix format. For instance, to decompose a weight of a fully-connected layer, $W_f$ with a coordinate format (COO), if $W_f$ does not contain any sparsity, COO decomposes $W_f$ with three times the number of parameters comparing to $W_f$. In the case of the convolutional layer with the weight, $W_c$, this overhead becomes worse. The number of parameters becomes five times comparing to $W_c$. With **FTTTF** model, the sparsity in each **T** layer is roughly 0.5. Using COO format consumes 1.5 times the amount of origin parameters in the case of the fully-connected layer. This becomes worse in the case of the convolutional layer that consumes 2.5 times the origin amount of parameters.

Our implementation of the MPWN is designed layer by layer. We also optimize the layer by placing directives into it. Fig 3 displays an overview of our MPWN design on an FPGA. Since quantization is applied to convolutional and fully-connected layers only, we evaluate the performance and apply directives in these layers only. In Fig 3, we defined our notation as follows: *Conv#N* indicates a convolutional layer, *Fully connected#N* indicates a fully-connected layer, *BN#N* indicates a batch normalization layer [34], *Flatten* indicates a rearranging layer to convert the shape activation to operate in the fully-connected layer, and *N* indicates the order of the weight layer.

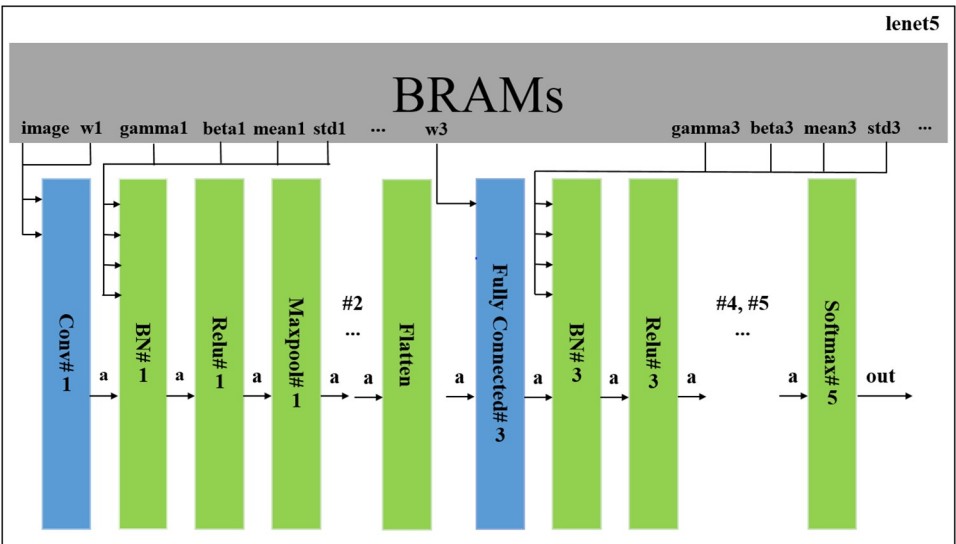

**Fig 3. Overview of mixed-precision weights network implementation.** All parameters of the MPWN are stored in BRAMs. Blue blocks indicate blocks that are optimized with directives, while green blocks indicate blocks that are not optimized with directives.

To explain how we place the directives, we must first define a convolutional and fully-connected layer. The convolutional layer is defined in Eq (13), where $W \in \mathbb{R}^{C_{out} \times C_{in} \times K_r \times K_c}$, $X \in \mathbb{R}^{C_{in} \times I_r \times I_c}$, $K_r$ denotes the number of kernel rows, $K_c$ denotes the number of kernel columns, $C_{in}$ denotes the number input channels, $C_{out}$ denotes the number of output channels, $I_r$ is the input activation row, $I_c$ is the input activation column, $O_r$ is the output feature row, $O_c$ is the output feature channel, and $F$ is the feature map. The convolutional layer operation consists of six for loops that can be accelerated with parallelism. It should be noted that the bias term is ignored because the convolutional layer is followed by batch normalization [34], which consists of a term that acts as a bias.

$$F = \sum_{n=1}^{C_{out}} \sum_{m=1}^{O_r} \sum_{l=1}^{O_c} \sum_{k=1}^{C_{in}} \sum_{j=1}^{K_r} \sum_{i=1}^{K_c} W_{n,k,j,i} X_{k,j+m,i+l} \tag{13}$$

In the convolutional layer function in VHLS, we utilize parallelism by placing the *UNROLL* directive inside the $O_c$, loop which hints that all loops below should be computed in parallel. In addition, we also place the *PIPELINE* directive inside the $O_r$ loop to further accelerate the operation that unrolled. In the convolutional layer, we apply *ARRAY_PARTITION* with a factor of 8 to both $W$ and $X$.

We define a fully-connected layer or matrix multiplication as Eq (14), where $W \in \mathbb{R}^{R \times C}$, $X \in \mathbb{R}^C$, $b \in \mathbb{R}^C$, $R$ denotes the number of rows of the weight, and $C$ denotes the number of columns of the weight.

$$F = \sum_{j=1}^{C} \sum_{i=1}^{R} W_{j,i} X_j + b_j \tag{14}$$

In the matrix multiplication function in VHLS, we place *PIPELINE* inside *C* for data pipelining of the process. We also apply *ARRAY_PARTITION* with a factor of 16 to both *W* and *X*. It should be noted that the first two fully-connected layers do not include the bias term *b*.

## Experimental results and discussion

In this section, we describe our experiments, which can be categorized into software simulation and hardware implementation.

### Software simulation

In this section, we separated into two sections: Fashion-MNIST and CIFAR10 datasets. In the Fashion-MNIST section, we further evaluated our heuristic rules by running a grid search covering all possible combinations of the MPWN with the LeNet-5 model. We applied BO to search for optimal MPWN combinations in terms of the *ASB* score. We evaluated how many search iterations were necessary to obtain a model with better *ASB* than heuristic rules. We also conducted an experiment to examine the effect of converting *half* to *float*. In the CIFAR10 section, we evaluated the robustness of our proposed methods by performing BO searches with ResNet-18 [35] model and CIFAR10 dataset.

**Fashion-MNIST.**   To evaluate the MPWN model, we used the Fashion-MNIST dataset [28] as a benchmark image dataset. Fashion-MNIST is a clothing image dataset that consists of 60,000 training images and 10,000 test images. Each image is a grayscale image consisting of 28x28 pixels. We preprocessed each pixel value to the range [0, 1] by dividing each image pixel by the maximum pixel intensity, 255. In this part, our MPWN model was programmed using PyTorch [36], a deep learning framework. The base structure of the CNN that we applied to the MPWN was LeNet-5 [4] with the structure: $6C5 - MP\,2 - 16C5 - MP2 - 120FC - 84FC - 10Softmax$, where *C5* is a $5 \times 5$ convolutional layer, *MP*2 is a $2 \times 2$ max-pooling layer, *FC* is a fully-connected layer and *Softmax* is an output layer. We used the rectified linear unit (ReLU) as the activation function, and applied dropout [30] in the fully-connected layers with $p = 0.5$ except in the last layer. We utilized batch normalization [34] to stabilize our training process, and our model was optimized with Adam [37] with an initial learning of $10^{-3}$. We trained all models for 200 epochs and stepped down the learning rate to one-tenth every 75 steps. We set the training batch size to 128 and we utilized *ASB* with $\alpha = 1, \beta = 1$, and $\gamma = 1$ in this experiment.

To visualize the heuristic rules, we performed a grid-search across all possible combinations of the MPWN; in other words, we trained $3^5 = 243$ combinations of the model. We summarized all metrics from these combinations into three box plots. In Figs 4 and 5. present boxplots that display the test accuracy on the y-axis and the type of weight layer on the x-axis. By running all possible combinations, **F** in the first and last layer correlates with the test accuracy compared with other weight layers. However, **F** in the last layer has an excessively high variance compared with **F** on the first layer. This reveals that we can update the last heuristic rules by changing the last layer of the CNN from **F** to other types of layers. However, the first should still remain **F**.

The third layer of LeNet-5 contains the highest number of parameters compared with other weight layers. From the first heuristic rule, by setting the third layer as **T** affects the sparsity of the model, as illustrated in Fig 6. Note that the **T** layer in **FBTBF** contains sparsity within the layer as 0.4974. This amount of sparsity can be counted as 0.3495 sparsity of the model. This first heuristic rule contains another advantage. Placing **T** only in layers that contain a large number amount of parameters (fully-connected layer) eases the hardware implementation relative to the convolutional layer, which contains a large number amount of for loops.

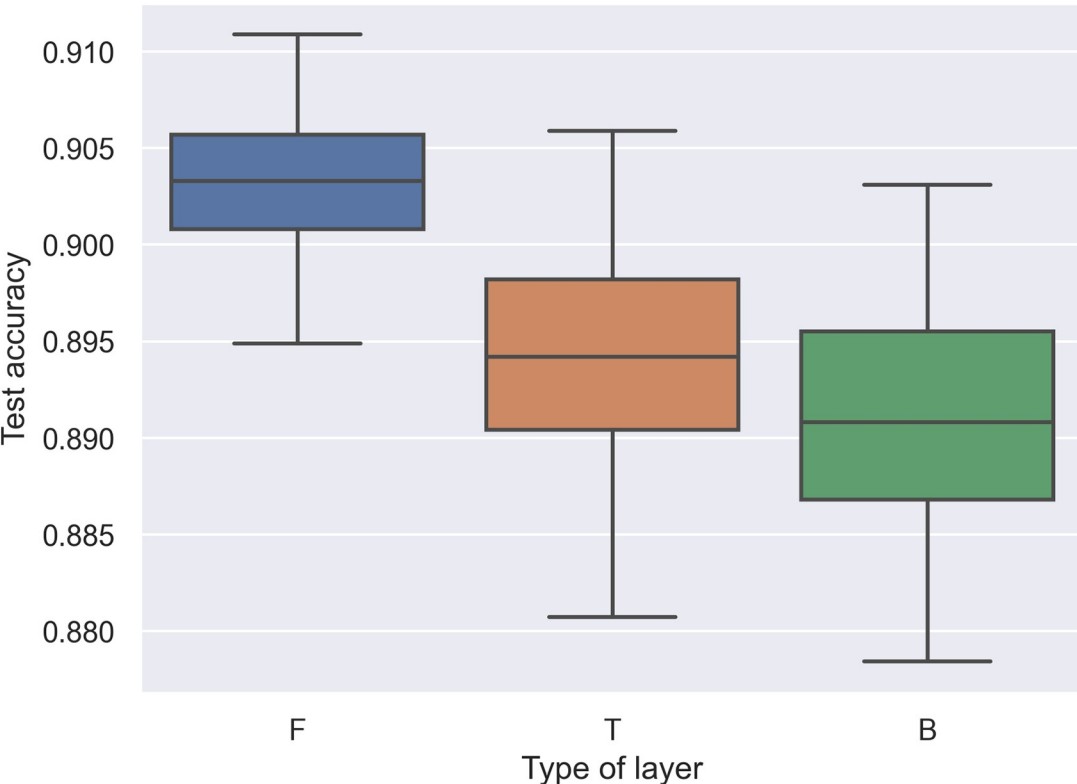

**Fig 4. Box plot of test accuracy and effect of layer type in the first layer.**

The results of MPWN, TWN, BC, and BNN are presented in Table 3. In Table 3, we defined the **ASB before** as *ASB* without the min-max normalization and we also defined **ASB** as *ASB* with the min-max normalization. We report metrics in the training epoch that achieved the highest test accuracy. It should be noted that to scale with other methods, we did not apply the scaling factor $S_i$ in Eq (5) to the TWN. We also did not clip weights in **B** layer with Eq (3). With heuristic rules, the **FBTBF** model was the optimized model. As displayed in Table 3, **FBTBF** obtained the advantages (i.e., accuracy, sparsity, and the number of bits) from the 16-bit floating-point, TWN, and BC models in a single model. However, by running all possible combinations of the MPWN, we found that the best *ASB* model without the min-max normalization was **FTTTT** with *ASB* as 0.7558. Comparing MPWN, TWN, and BC with BNN, BNN with binary activations promises a better in terms of hardware friendliness where its feature maps accumulation can be replaced with popcount operation. However, its test accuracy is significantly dropped comparing with other methods.

We plotted a box-plot of distributions of each element in *ASB* score as illustrated in Fig 7. In Fig 7, there are differences in mean and variance between each metric. Therefore, each metric contributes differently to *ASB*. We applied the Pearson correlation to measure the contributions of *a*, *s*, and *b* to *ASB*. We found that the Pearson correlation between *a*, *s*, and *b* to *ASB* are -0.2249, 0.5575, and 0.8493, respectively. The main issue in this *ASB* is the correlation between *a* and *ASB* is negative. We expected this issue is caused from the variance of *a* is insignificant or $5.36 \times 10^{-5}$ comparing with *s* and $b_{nor}$ that is 0.09865 and 0.03881, respectively.

One of main contribution of the small variance of *a* is *a* contains a narrow range of distribution from the minimum value at 0.8784 and maximum value at 0.9109. To improve the

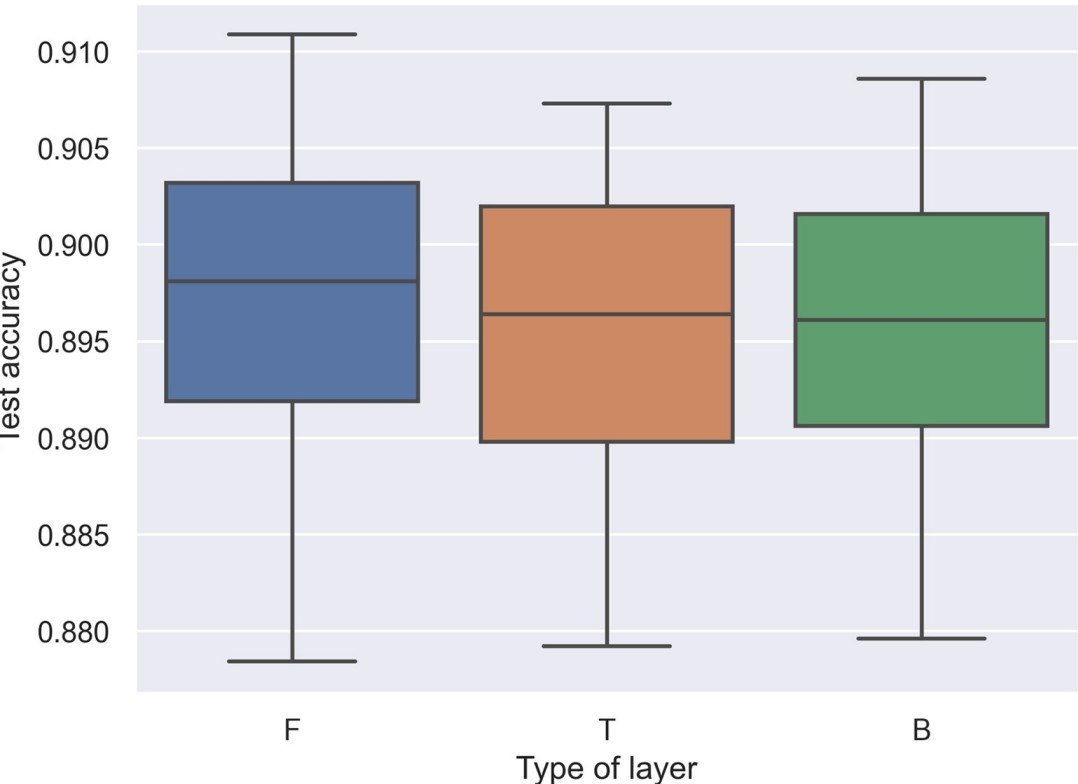

**Fig 5. Box plot of test accuracy and effect of layer type in the last or fifth layer.**

variance issue of $a$, we can rescale all element in $ASB$ into the same scale by using a min-max normalization. By applying with the min-max normalization, normalized metrics are illustrated in Fig 8. The Pearson correlation between $a$, $s$, and $b_{nor}$ to $ASB$ became 0.2816, 0.7743, and 0.5153, respectively and the variance of $a$, $s$, and $b_{nor}$ becomes 0.05074, 0.1302, and 0.0112, respectively. The best combination in term of $ASB$ is changed from **FTTTT** to **FTTTF**. We expected by rescaling the range of $a$, this signified the correlation between the **F** layer to $ASB$ score, therefore the **FTTTF** becomes more important than **FTTTT**.

The min-max normalization requires minimum and maximum values of $a$, $s$, and $b_{nor}$ to perform the normalization. However, to find actual minimum and maximum values, in the worst case, this requires training all possible MPWN combination which is not feasible with the large model. Therefore, we estimated the minimum and maximum values by using the estimating rules that we mentioned in the mixed-precision weights network section instead. To evaluate the estimating rules, we compared the mean square error (MSE) between normalized values from the estimating rules to the normalized values from the actual maximum and minimum values. We also provided a comparison with normalized values from maximum and minimum values that were known from the random search. These comparisons are shown in Table 4. The box-plot of $ASB$ after min-max normalization with actual and estimated values are illustrated in Fig 8.

From Table 4, by using the random search for 50 iterations or approximately one-fifth of all possible combinations, the random search achieved the lower MSE of $ASB$ comparing with the estimating rules or **Proposed**. These results indicate that our estimating rules did not provide

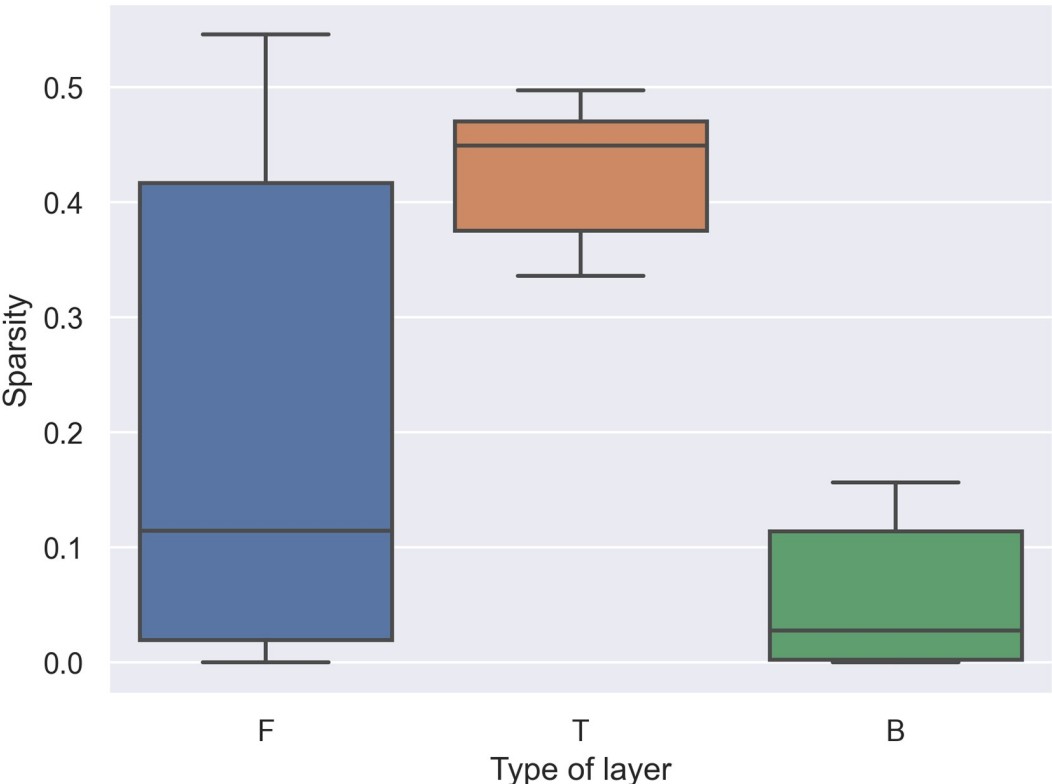

**Fig 6. Box plot of sparsity and effect of layer type in the third layer.**

the perfect estimations of the minimum and maximum values. However, our estimating rules still provide a better alternative if we do not wish to train for 50 different models.

**Bayesian optimization.** We conducted an experiment with BO and the *ASB* score with $\alpha$ = 1, $\beta$ = 1, and $\gamma$ = 1. We evaluated the effectiveness of BO in finding optimal combinations of the MPWN. We applied BO from an implementation from `hyperopt` [38], and used BO to search for optimized MPWN models for 100 iterations. That is, we searched with BO for 100 different combinations of the MPWN. In each iteration, BO received feedback or the *ASB* score and learned what the subsequent combination of MPWN should be. Fig 9 presents the *ASB* score improvement of BO in each iteration. The *ASB* score on the y-axis changed only

**Table 3. Comparison between different combinations the mixed-precision weights network.** ASB before denotes *ASB* without the min-max normalization and ASB denotes *ASB* with the min-max normalization.

| Type of layers | Test accuracy | Sparsity | Amount of bit | ASB before | ASB |
|---|---|---|---|---|---|
| Full Precision | **0.9109** | 0.0 | 707,040 | 0.3036 | 0.3333 |
| TWN [11] | 0.8928 | 0.4852 | 88,380 | 0.751 | 0.7551 |
| BC [9] | 0.8798 | 0.0 | **44,190** | 0.6057 | 0.3477 |
| BNN [10] | 0.8475 | 0.0 | **44,190** | 0.595 | 0.0164 |
| **FBTBF** | 0.8984 | 0.3495 | 89,760 | 0.7069 | 0.7289 |
| **FTTTT** | 0.9036 | **0.4919** | 90,480 | **0.7558** | 0.8689 |
| **FTTTF** | 0.9071 | 0.4911 | 102,240 | 0.7512 | **0.8984** |

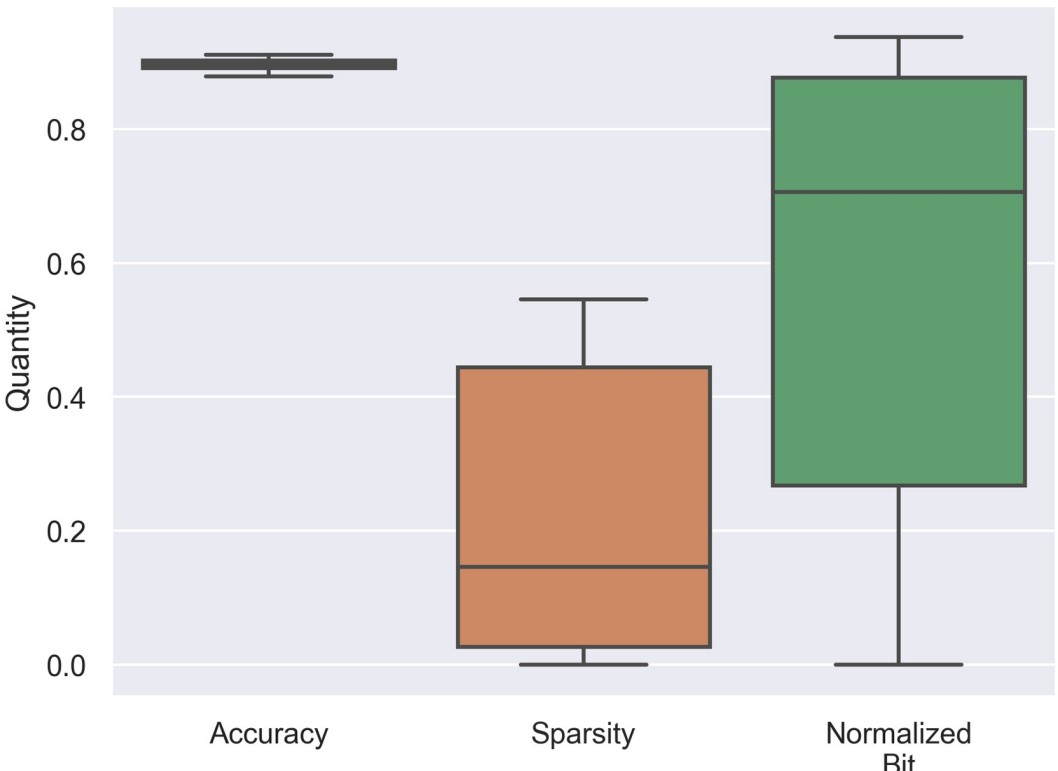

**Fig 7. Distributions of accuracy, sparsity, and normalized bit from all possible combinations of MPWN with LeNet-5.**

when BO found a new combination that produced a higher *ASB* score. We determined that BO was able to find the global maximum **FTTTF** after 79 iterations (i.e., approximately one-third of all possible search numbers). BO was able to find a better alternative to **FBTBF** or a model with an *ASB* score higher than 0.7289 after 5 iterations of searching. Therefore, heuristic rules still provide a better alternative if we do not wish to spend the time and resources to train 5 different models.

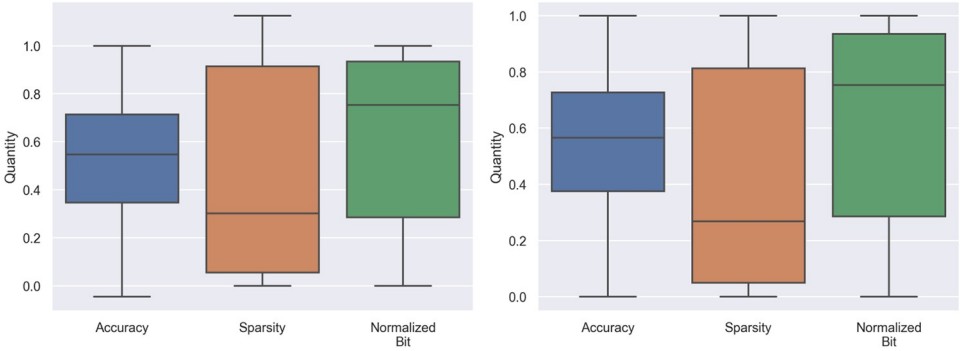

**Fig 8. Box plots of each elements in *ASB* after the min-max normalization.** Left: after the min-max normalization with estimated minimum and maximum values. Right: after the min-max normalization with actual minimum and maximum values.

**Table 4. Comparison between the min-max normalization from the random search and the estimating rules.** Proposed denotes the minimum and maximum values from the estimating rules. GPU time indicates the total training time with the same setting as the Fashion-MNIST section using NVIDIA GeForce GTX 1080 and Intel Xeon CPU E5-1620 v3.

| Round of random search | MSE of a | MSE of s | MSE of b | MSE of $ASB$ | GPU time (minute) |
|---|---|---|---|---|---|
| 10 | 0.0251 | 0.0119 | $1.377 \times 10^{-3}$ | $4.496 \times 10^{-3}$ | 208 |
| 30 | $2.71 \times 10^{-3}$ | $3.307 \times 10^{-3}$ | $0.2697 \times 10^{-3}$ | $0.6016 \times 10^{-3}$ | 603 |
| 50 | $2.01 \times 10^{-3}$ | **0.0** | $2.769 \times 10^{-6}$ | **$0.211 \times 10^{-3}$** | 991 |
| **Proposed (3)** | **$0.524 \times 10^{-3}$** | $4.827 \times 10^{-3}$ | **0.0** | $0.366 \times 10^{-3}$ | **59** |

**Effect of *float* and *half* on MPWN model.** After training the **FTTTF** model for 200 epochs, we measured the effect of converting all parameters in the MPWN from *float* to *half*. We conducted this experiment in the PyTorch environment. The results are presented in Table 5. We observed that the performance of the **FTTTF** does not change. Therefore, in this case, we were able to convert *float* to *half* without performance loss.

**CIFAR10.** In this section, we further evaluated our proposed methods by applying BO with ResNet-18 [35] and CIFAR10 [29] dataset. ResNet-18 is a CNN that consists of 18 weight layers with several residual connections which allow both feature maps and gradients to flow with. CIFAR10 is an image dataset that consists of 50,000 training images and 10,000 test images. Each image is an RGB image consisting of 32x32 pixels. We preprocessed each image in channel-wise direction with means and standard deviations of the training dataset. To

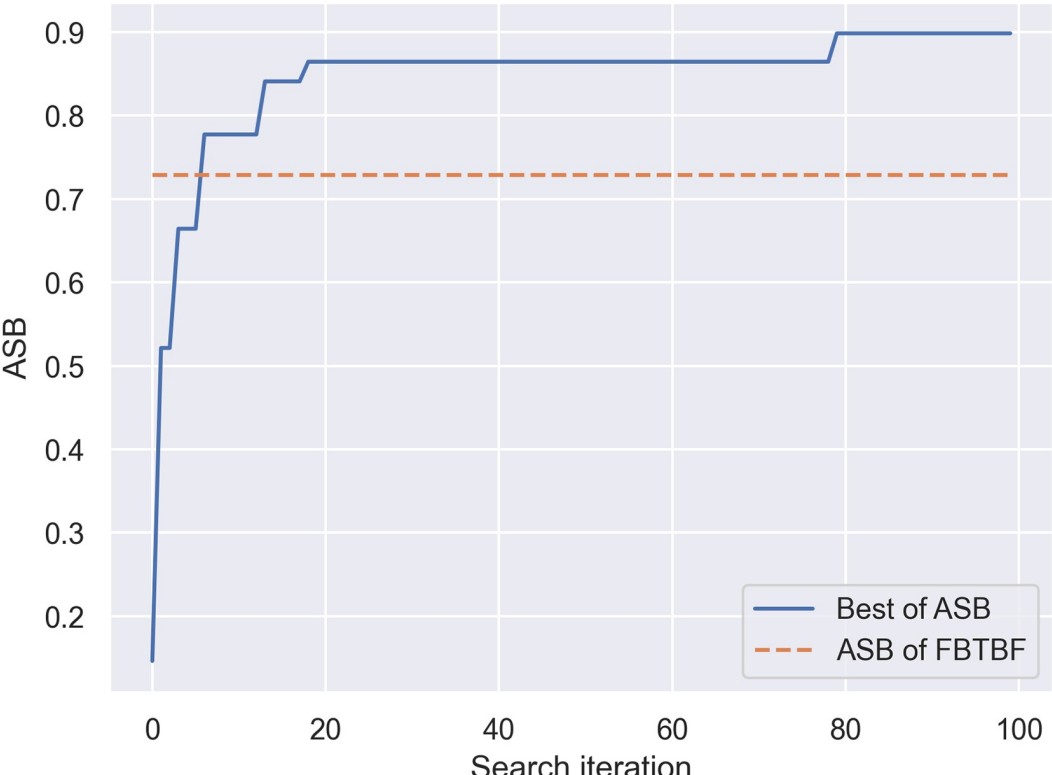

**Fig 9. Bayesian optimization search with ASB score.** This graph displays the best $ASB$ in the current iteration search. The score changes when a higher score is found. The orange dashed line indicates the normalized $ASB$ score of the heuristic rule (0.7289).

**Table 5. Comparison between *float* and *half* from FTTTF model.**

|  | Test accuracy | Difference with *float* |
|---|---|---|
| *float* | 0.9053 | 0.0 |
| *half* | 0.9053 | 0.0 |

modify ResNet-18 to operate with CIFAR10, we adjusted a first convolutional layer of ResNet-18 to 64C3 with stride of one and we also removed the max-pooling layer. To avoid overfitting, we applied data augmentations by random padding border pixels of an image with four pixels and random crop the image back to original 32x32 pixels. The image was further augmented by random horizontal flipping. We trained the ResNet-18 with 256 batch size for 150 epochs with Adam [37]. We set an initial learning rate with $10^{-3}$ and step down to one-tenth every 50 epochs. We quantized all weight layers within ResNet-18, including weights from residual connections. For the notation, We denote weights from residual connections after the underscore. For instance, **FFFFF_FFF** indicates a neural network with five **F** weight layers and three residual connections with **F** weights.

We performed 70 rounds of BO search with *ASB* ($\alpha = 1$, $\beta = 1$, and $\gamma = 1$). Each metric of *ASB* was normalized with the estimating rules. Since there are $3^{18}$ possible combinations of MPWN, outcomes of this BO search are not expected to contain a global maximum of *ASB*. We displayed the top-5 *ASB* combinations from BO search in **Proposed** section of Table 6. We also included a **Baseline** section that consists **F**, **B**, and **T** models. Note that in **Baseline**, we also calculated *ASB* with the estimating rules.

In Table 6, we discovered patterns of heuristic rules in the top-5 combinations. In ResNet-18, the number of weight parameters increases gradually from the first to last convolutional layer. Therefore, the last three convolutional layers of ResNet-18 contribute 63.34 percent of all weight parameters. Using the first heuristic rule (layers that contain a large number of weight parameters should be **T**), these last three convolutional layers should be **T** to maximize the sparsity. This pattern of first and third heuristic rule (the first and last layers should be **F**) can be identified throughout **Proposed**. With patterns of heuristic rules appeared in top-5 combinations, this signified a correlation between the heuristic rules and optimal *ASB* combinations.

Using BO with *ASB* allows searching a better alternative to models from **Baseline**. For example, **FFFTTTTFTBBBFTTTTTF_TBT** or the top-1 model in **Proposed** contains all desired properties of *ASB* or high test accuracy, sparsity while maintaining a low amount of bits in a single model.

**Table 6. Top-5 combinations from each BO search.** Iteration denotes the number of BO searches. Note that the *a*, *s*, and *b* are not normalized with the min-max normalization and Iteration starts with 0.

|  | Combinations | *ASB* | *a* | *s* | *b* | Iteration |
|---|---|---|---|---|---|---|
| **Baseline** | **FFFFFFFFFFFFFFFFFF_FFF** | 0.3333 | **0.9366** | $3.584 \times 10^{-6}$ | 178,629,632 | - |
|  | **TTTTTTTTTTTTTTTTTT_TTT** | **0.7852** | 0.9221 | **0.93557** | 22,328,704 | - |
|  | **BBBBBBBBBBBBBBBBBB_BBB** | 0.3333 | 0.9209 | $6.27 \times 10^{-7}$ | **11,164,352** | - |
| **Proposed** | **FFFTTTTFTBBBFTTTTTF_TBT** | **0.8156** | 0.9276 | 0.8086 | 32,713,728 | **64** |
|  | **FFTBTTFTFBFFTTBTTF_TBT** | 0.8092 | **0.935** | 0.5996 | 40,860,672 | 37 |
|  | **BFFTFTFBFBTFTTTTTF_TBT** | 0.7915 | 0.9248 | **0.8486** | 35,858,112 | 4 |
|  | **FFFTTTFFBBTFTTTTTF_TBB** | 0.7717 | 0.9232 | 0.8413 | 35,236,864 | 22 |
|  | **FFTTTTFBFBTFTTTBTF_TTB** | 0.7504 | 0.9286 | 0.5945 | **32,394,240** | 55 |

## Hardware synthesis and implementation

In this section, we synthesized a **FTTTF** model with Vivado HLS version 2019.02 and implemented this model with Xilinx Vivado 2019.2 [39]. We evaluated the model in terms of latency and hardware area compared with a conventional 32-bit floating-point model. We also performed comparisons between models with directives and without directives.

All hardware synthesis results were obtained by using VHLS with the option of C synthesis. The target FPGA was Zynq UltraScale+ MPSoC ZCU102 (ZCU102) or xczu9eg-ffvb1156-2-i. ZCU102 consisted of the following hardware resources: 1824 BRAM_18K, 2520 DSP48E, 548,160 FF, and 274,080 LUT. Our FPGA operated with a target clock frequency of 100 MHz, and our implemented model used weights and biases from training in the software simulation part.

**XOR signed-bit and ternary bitwise operation synthesis.** We synthesized XSB and TBO as a replacement for multiplication between *half* and *int1* and *half* and *int2*, respectively. Therefore, we compared hardware resources and latency with multiplication between other data types, as illustrated in Table 7.

Using XSB instead of floating-point arithmetic significantly reduced the latency and hardware resources. XSB consumed only two LUTs to perform multiplication, while TBO consumes 32 LUTs. Each pair of LUTs used to construct a logic gate. Therefore, XSB utilizes a logic gate, while TBO utilizes 16 logic gates. Note that in both XSB and TBO, the latency cannot be zero in practice. We only displayed the results from VHLS C synthesis.

In term of behavior, XSB should perform in the same manner as Listing 3. Listing 3 detects the most significant bit (MSE) of the binary weight and flips the MSE of *float*. However, detecting requires a control logic which is expensive VHLS. For example, by using Listing 3 to C synthesis with Vivado HLS 2019.2, it consumes 2 cycles at 100 MHz, 2 DSP48Es, 97 FFs, and 150 LUTs. Comparing with XSB that consume only 2 LUTs and 0 latency, XSB is more efficient in both hardware utilization and latency.

**Listing 3. XOR signed-bits using a control logic**.

```
void xor_signed_bit_using_control_logic(int1 w, half x, half &o)
{
    #pragma HLS INLINE OFF
    o = w.sign() == 0 ? x: −x;
}
```

**Hardware synthesis.** We synthesized the MPWN with **FTTTF** into an FPGA with VHLS. We performed several comparisons with the model with *float* and with and without directives. We used the following notations. **Proposed** signifies that all *float* data type were replaced with *half*; in **T**, the multiplication was replaced with TBO. **Base-line** signifies that all data type were *float* and that all operations in the model were the floating-point arithmetic. **Directive** signifies

**Table 7. Comparison of latency and hardware utilization of multiplication between two variables with data type 1 and 2, respectively.** The latency unit is a clock cycle. The two last row represents XOR signed bit and ternary bitwise operation, respectively.

| Data type 1 | Data type 2 | Latency | Hardware Resource | | |
|---|---|---|---|---|---|
| | | | DSP48E | FF | LUT |
| *double* | *double* | 4 | 11 | 304 | 236 |
| *float* | *float* | 1 | 3 | 130 | 150 |
| *half* | *half* | 1 | 2 | 66 | 49 |
| **half** | **int1** | **0** | **0** | **0** | **2** |
| **half** | **int2** | **0** | **0** | **0** | **32** |

**Table 8. Comparison between different FPGA synthesis of LeNet-5 layer by layer in terms of latency (ms).**

| Layer | Base-line | Proposed | Base-line directive | Proposed directive | CPU ARM Cortex-A53 | Comparing Proposed directive and CPU |
|---|---|---|---|---|---|---|
| *Conv#1* | 7.364 | 4.738 | 0.027 | **0.0264** | 0.148±0.009 | 5.61x |
| *Conv#2* | 13.059 | 5.369 | 0.122 | **0.119** | 0.684±0.009 | 5.75x |
| *Fully connected#3* | 2.460 | 0.924 | 0.020 | **0.0147** | 0.173±0.002 | 11.77x |
| *Fully connected#4* | 0.808 | 0.304 | 0.009 | **0.006** | 0.066±0.006 | 11.0x |
| *Fully connected#5* | 0.067 | 0.042 | 0.004 | **0.002** | 0.004±0.0 | 2.0x |

that we applied optimization directives in VHLS to optimize the latency of the model with parallelism. However, this resulted in a trade-off of higher hardware utilization.

A comparison of the latency and resources using these methods is presented Tables 8 and 9, respectively. Table 8 includes comparisons with ARM Cortex-A53. We utilized ARM Cortex-A53 in Zynq UltraScale+ MPSoC ZCU102. To use this CPU, we generated a PetaLinux image [40] and executable files from C++ files using SDSoC 2018.3 [41]. The setting of these C++ files was the same as **Base-line**. We ran executable files three times and reported the latency mean and interval of two standard deviations.

We observed that the convolutional layers had a longer latency than the fully-connected layers even though there were fewer parameters. We hypothesized that this was due to the complexity of the convolutional layer, and the VHLS performed worse when dealing with a

**Table 9. Comparison between different FPGA synthesis in terms of hardware utilization.** The number inside parentheses indicates the percentage of hardware utilization of Zynq UltraScale+ MPSoC ZCU102. In the *Total* row, some layers may not be included, such as the flatten, max-pooling, and batch normalization layers.

| Layer | Components | Base-line | Proposed | Base-line directive | Proposed directive |
|---|---|---|---|---|---|
| *Conv#1* | BRAM-18K | 0 (0%) | 0 (0%) | 0 (0%) | 0 (0%) |
| | DSP48E | 5 (0%) | **4 (0%)** | 170 (7%) | **136 (5%)** |
| | FF | 591 (0%) | **317 (0%)** | 71,343 (13%) | **24,194 (4%)** |
| | LUT | 814 (0%) | **589 (0%)** | 45,469 (17%) | **24,858 (9%)** |
| *Conv#2* | BRAM-18K | 0 (0%) | 0 (0%) | 0 (0%) | 0 (0%) |
| | DSP48E | 5 (0%) | **2 (0%)** | 70 (3%) | **28 (1%)** |
| | FF | 598 (0%) | **254 (0%)** | 120,868 (22%) | **47,089 (8%)** |
| | LUT | 894 (0%) | **668 (0%)** | 69,094 (25%) | **43,888 (16%)** |
| *Fully connected#3* | BRAM-18K | 0 (0%) | 0 (0%) | 0 (0%) | 0 (0%) |
| | DSP48E | 5 (0%) | **2 (0%)** | 160 (6%) | **64 (2%)** |
| | FF | 542 (0%) | **161 (0)** | 44,342 (8%) | **24,294 (4%)** |
| | LUT | 538 (0%) | **302 (0%)** | 38,933 (14%) | **21,869 (8%)** |
| *Fully connected#4* | BRAM-18K | 0 (0%) | 0 (0%) | 0 (0%) | 0 (0%) |
| | DSP48E | 5 (0%) | **2 (0%)** | 150 (6%) | **60 (2%)** |
| | FF | 566 (0%) | **185 (0%)** | 31,682 (6%) | **13,486 (2%)** |
| | LUT | 544 (0%) | **302 (0%)** | 27,134 (10%) | **14,874 (5%)** |
| *Fully connected#5* | BRAM-18K | 0 (0%) | 0 (0%) | 0 (0%) | 0 (0%) |
| | DSP48E | 5 (0%) | **4 (0%)** | 140 (6%) | **112 (4%)** |
| | FF | 526 (0%) | **227 (0%)** | 26,240 (5%) | **12,891 (2%)** |
| | LUT | 530 (0%) | **304 (0%)** | 19,937 (7%) | **11,570 (4%)** |
| *Total* | BRAM-18K | 40 (2%) | **25 (1%)** | 51 (3%) | **29 (1%)** |
| | DSP48E | 54 (2%) | **43 (1%)** | 719 (29%) | **429 (17%)** |
| | FF | 8,228 (2%) | **3,831 (0%)** | 302,914 (55%) | **126,198 (23%)** |
| | LUT | 14,360 (5%) | **9,232 (3%)** | 214,386 (78%) | **126,787 (46%)** |

**Table 10. Comparison between baseline implementation of LeNet-5, our proposed method, and related works in term of latency.**

|  | Latency (ms) | Comparison with Baseline |
|---|---|---|
| **Baseline** | 24.799 | 1x |
| **Proposed** | 11.995 | 2.07x |
| **Baseline directive** | 1.222 | 20.29x |
| **Proposed directive** | **0.786** | **31.55x** |
| **Rongshi et al.** [26] | 9.135 | 2.715x |
| **Cho et al.** [27] | 3.581 | 6.925x |
| **GUINNESS** [25] | 7.882 | 3.146x |

large number of for loops in the convolutional layer. With the directives, both **Proposed** and **Base-line** significantly improved the latency; however, they also significantly increased the hardware utilization. Compared with ARM Cortex-A53, we were able to reduce the latency 2.0 to 11.77 times depending on the type of layer.

Table 10 includes comparisons in term of latency of MPWNs, Rongshi et al. [26], GUIN-NESS [25], and Cho et al. [27]. Rongshi et al. proposed a 32-bit floating-point LeNet-5 on Xilinx Zybo Z7 board (zynq7020). Cho et al. proposed a fixed-point LeNet-5 model that targets xczu9eg-ffvb1156-2-i. Cho et al. utilized 20-bit fixed-point on the first layer and 16-bit fixed-point on the latter layers. GUINNESS is a graphical user interface for training BNN on a GPU and deploying BNN on an FPGA. We utilized the GUINNESS from [42] to construct a BNN with a default LeNet-5 configuration of GUINNESS. This BNN was set to target Zynq Ultra-Scale+ MPSoC ZCU102 (xczu9eg-ffvb1156-2-i). All of the related works operate with the same 100 MHz frequency.

Note that there are serveral differences between our and related models. The first difference is Rongshi et al. and Cho et al. applied the third layer as a convolutional layer instead of a fully-connected layer that we utilized. The second one is the default setting of GUINNESS for LeNet-5 is $64C3 - 64C3 - 64C3 - 32AP - 10FC$, where $32AP$ is a global average pooling that averages feature maps in width and height directions. The third one is Rongshi et al., and Cho et al. did not apply batch normalization layers. The fourth difference is Cho et al. replaced max-pooling layers with average pooling layers and utilized a Tanh activation function instead of ReLU. The last one is GUINNESS and Cho et al. expect grayscale 32x32 images as inputs instead of grayscale 28x28 images that we used. Our **Proposed directive** performed 11.62, 10.03, and 4.556 times less latency comparing Rongshi et al., GUINNESS, and Cho et al., respectively.

**Hardware implementation.** To implement the model, we exported our model in VHLS as intellectual property (IP). We used Vivado 2019.2 to implement the exported IP to ZCU102. The results of the implementation are provided in Tables 11 and 12. In Table 11, there is an additional hardware resource which is the look-up table random-access memory (LUTRAM). Comparing Tables 9 and 11, the synthesis and implementation displayed a different amount of hardware utilization. Note that in Table 11, Cho et al. did not provide the hardware utilization from the implementation. Therefore, we reported the Cho et al. synthesis results in Table 11. In terms of **Proposed** and **Proposed directive**, the implementation exhibited significantly reduced hardware utilization than the synthesis. Our **Proposed** reduced LUT 2.31 times, LUTRAM 11.25 times, FF 2.89 times, BRAM 1.6 times, and DSP 1.25 times compared to **Base-line**. **Proposed directive** further reduced LUT 2.59 times, LUTRAM 4.89 times, FF 2.92 times, BRAM 1.76 times, and DSP 1.68 times compared with **Base-line directive**. Comparing with

**Table 11. Comparison between FPGA implementations of LeNet-5 in the term of hardware utilization.** In an improvement factor column displays pairwise comparisons between Baseline with Proposed and Baseline directives with Proposed directives. All improvement factors from related works are compared with **Baseline directives**.

| | Components | Total amount | Improvement factor |
|---|---|---|---|
| **Baseline** | LUT | 8,305 | 1.0x |
| | LUTRAM | 180 | 1.0x |
| | FF | 6,787 | 1.0x |
| | BRAM | 20 | 1.0x |
| | DSP | 55 | 1.0x |
| **Proposed** | LUT | **3,599** | **2.31x** |
| | LUTRAM | **16** | **11.25x** |
| | FF | **2,345** | **2.89x** |
| | BRAM | **12.5** | **1.6x** |
| | DSP | **44** | **1.25x** |
| **Baseline directive** | LUT | 209,720 | 1.0x |
| | LUTRAM | 49,477 | 1.0x |
| | FF | 243,540 | 1.0x |
| | BRAM | 25.5 | 1.0x |
| | DSP | 719 | 1.0x |
| **Proposed directive** | LUT | 81,050 | 2.59x |
| | LUTRAM | 10,142 | 4.88x |
| | FF | 83,266 | 2.92x |
| | BRAM | **14.50** | **1.76x** |
| | DSP | 429 | 1.68x |
| **Rongshi et al. [26]** | LUT | 14,659 | 14.31x |
| | FF | 14,172 | 17.18x |
| | BRAM | 119.5 | 0.2134x |
| | DSP | **125** | **5.752x** |
| **Cho et al. [27]** | LUT | 32,589 | 6.435x |
| | FF | 33,585 | 7.251x |
| | BRAM | 95 | 0.2684x |
| | DSP | 143 | 5.028x |
| **GUINNESS [25]** | LUT | **5,034** | **41.66x** |
| | LUTRAM | **278** | **178x** |
| | FF | **4,417** | **55.14x** |
| | BRAM | 23.5 | 1.085x |

**Table 12. Comparison between implementations of LeNet-5 in terms of total on-chip power (W).** The improvement factor column displays a pairwise comparison between Baseline with Proposed and Baseline directives with Proposed directives. All improvement factors from related works are compared with Baseline directives.

| | Total on-chip power (W) | Improvement factor |
|---|---|---|
| **Baseline** | 0.852 | 1.0x |
| **Proposed** | **0.72** | **1.18x** |
| **Baseline directive** | 2.901 | 1.0x |
| **Proposed directive** | 1.414 | 2.05x |
| **Rongshi et al. [26]** | 1.8 | 1.612x |
| **Cho et al. [27]** | - | - |
| **GUINNESS [25]** | **0.66** | **4.395x** |

Rongshi et al., GUINNESS, and Cho et al., our **Proposed directive** utilizes more hardware utilization except for BRAMs.

Table 12 presents a comparison in terms of power utilization. **Proposed** reduced the power consumption of **Baseline** 1.18 times. However, **Proposed directive** further reduced the power consumption 2.05 times compared to **Base-line directive**. Although our **Proposed directive** consumed more hardware resources than Rongshi et al., our power consumption of **Proposed directive** is less than Rongshi et al. 1.27 times. However, our **Proposed directive** still consumes 2.142 times more power consumption than GUINNESS.

## Conclusion

In this study, we introduced MPWN, a QNN that jointly utilizes three weight spaces: floating-point, binary, and ternary. We proposed a systematized search to find optimal MPWN combinations with BO and the *ASB* score. To ensure each metric of *ASB* contains a positive correlation with *ASB*, we introduced a min-max normalization to rescale each metric of *ASB*. To accelerate the min-max normalization process, we provided estimating rules to estimate minimum and maxmum values of each *ASB* metric using information from only three models. We further evaluated previously proposed heuristic rules and the trade-off between heuristic rules and BO search. Our hardware implementation of the MPWN exploited the weight space in the MPWN with TBO and a specific data type. All of these elements demonstrated that the MPWN can be implemented in an FPGA with significantly fewer hardware resources and lower on-chip power consumption, and latency than a conventional 32-bit floating-point neural network.

## Author Contributions

**Conceptualization:** Ninnart Fuengfusin.

**Formal analysis:** Ninnart Fuengfusin.

**Funding acquisition:** Hakaru Tamukoh.

**Investigation:** Ninnart Fuengfusin.

**Methodology:** Ninnart Fuengfusin.

**Project administration:** Hakaru Tamukoh.

**Software:** Ninnart Fuengfusin.

**Supervision:** Hakaru Tamukoh.

**Validation:** Ninnart Fuengfusin, Hakaru Tamukoh.

**Writing – original draft:** Ninnart Fuengfusin.

**Writing – review & editing:** Ninnart Fuengfusin, Hakaru Tamukoh.

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
