## [Decision Letter · Decision Letter 0]

21 Dec 2020

PONE-D-20-36981

Mixed-precision weights network for field-programmable gate array

PLOS ONE

Dear Dr. Fuengfusin,

Thank you for submitting your manuscript to PLOS ONE. After careful consideration, we feel that it has merit but does not fully meet PLOS ONE’s publication criteria as it currently stands. Therefore, we invite you to submit a revised version of the manuscript that addresses the points raised during the review process.

We look forward to receiving your revised manuscript.

Kind regards,

Chi-Hua Chen, Ph.D.

Academic Editor

PLOS ONE

Reviewers' comments:

Reviewer's Responses to Questions

**Comments to the Author**

1. Is the manuscript technically sound, and do the data support the conclusions?

Reviewer #1: Partly

Reviewer #2: Yes

Reviewer #3: Yes

Reviewer #4: No

2. Has the statistical analysis been performed appropriately and rigorously? 

Reviewer #1: N/A

Reviewer #2: Yes

Reviewer #3: Yes

Reviewer #4: No

3. Have the authors made all data underlying the findings in their manuscript fully available?

Reviewer #1: Yes

Reviewer #2: Yes

Reviewer #3: No

Reviewer #4: Yes

4. Is the manuscript presented in an intelligible fashion and written in standard English?

Reviewer #1: Yes

Reviewer #2: Yes

Reviewer #3: Yes

Reviewer #4: Yes

5. Review Comments to the Author

Reviewer #1: This paper demonstrates an FPGA implementation of a mixed-precision inference design for convolutional neural networks. The authors synthesized a design on an FPGA and reports the results here. The scheme to perform mixed-precision inference was previously reported at the International Conference on Neural Information Processing 2018 by the same authors. Although the use of mixed-precision operations for inference is an interesting idea on its own, and the use of binary and ternary neural networks lands themselves naturally as optimized hardware implementations, this paper fails to present any scientific insight and data analysis to support the usefulness and novelty of the idea.

This paper covers 2 ideas: the idea of quantizing neural networks with layer-optimized mixed-precision weights and their hardware implementations on FPGAs. By using a simple Bayesian optimization scheme, the authors illustrate how the use of different quantization schemes on different layers of a neural network model would affect the final accuracy of the model. They demonstrate that in their specific example of inference using LeNet-5 on the Fashion-MINST dataset, the FTTTT configuration results in better accuracy as the heuristic model FBTBF, where F,T,B represents full precision, ternary quantization, and binary quantization respectively. Unfortunately, the analysis stops short with just one example of small network operating on one benchmark. A comprehensive analysis, especially on larger and more complex networks such as the ResNet family, would be needed to provide a better understanding of the true value of the proposed scheme.

Furthermore, in terms of their FPGA implementation, the paper presents the implementation on the FBTBF model instead of the FTTTT model that was just presented as superior. The authors spend a considerable amount of coverage discussing many low-level hardware properties of operating on binary and ternary networks. However, these are all widely understood standard techniques and properties of these highly quantized network. For example, the authors devote a long discussion on “XNOR signed-bit”, which involves multiplying a floating-point activation with a binary weight. At the heart of this multiplication, essentially it is a simple implementation of changing the sign of a value by flipping the MSB bit of a floating-point number depending on the multiplied binary weight. A smarter synthesis tools could have easily performed that operation when instructed to multiply by -1. Therefore, the significance of the long discussion on how bypassing current limitation of one specific high-level synthesis tools is very limited.

Finally, I find it disappointing in the paper when the study of mixed-precision model and their corresponding FPGA implementations are not coherently presented in the paper. After the long discussion on the BO algorithm, the authors claim, without fully justifying the underlying causes, that since “achieving an efficient implementation of the T layer in a convolutional layer is difficult”, they implement the heuristic model FBTBF on FPGA instead. Without proper hardware evaluation, it is difficult to justify for the benefit of the proposed BO method for finding mixed-precision network model. It also undermines the coherence of the paper. I suggest the authors to focus either on the optimization algorithms with more comprehensive network structures and datasets, or the authors may focus on the hardware implementations of different combination of network layers and show how they would have made the design more efficient than a fixed-precision network.

Reviewer #2: The article presenta a nice work of using mixed-precision weights networs (MPWN) to implement a quantization neral network.

The approach is well described, as well the software simulation and hardware implementation.

Reviewer #3: The authors need to compare their results with other works in the literature. There are many other compression approaches based on quantization applied to the Deep Neural Networks in the literature. Which does the advantages of the proposal presented in this manuscript?

The authors used a High-Level Synthesis (HLS) tool to implement the proposal on FPGA. However, these kinds of tools insert a high overhead resource in the generated hardware. The authors must explain more about that. Does the HLS decrease the results performance? Does it not better to implement without HLS?

Reviewer #4: 1. This paper presents a Mixed Precision Weight Network and its FPGA implementation by using binary weights, ternary weights and floating point weights. The paper is well written but lack in research novelty and research contribution. Please see comments below.

2. There are a number of papers in computer architecture and FPGA conferences as well as in ML/DL conferences/journals that deal with mixed precision networks. However, the current paper has no discussion of such prior work. Hence, it does not set the background for the current work properly. There is no comparison, qualitative and quantitative with such prior works either which is important to see where lies novelty in the current paper.

3. The motivation for using binary, ternary and FP weights in a neural network is not clear. It seems that the authors have based their decision based on their prior work [19]. However, the heuristics based prior work [19] does not explain how "large" or "small" is to be determined in the context of the current paper.

4. ASB is defined to be = (a+s+bnor)/3. However, the three quantities are not dimensionally equivalent. For instance, how is sparsity defined as a number - that should be explained and how dimensional analysis considerations are taken into account in this formula should be explained.

5. FPGA implementation described in this paper is regular textbook implementation. Unfortunately, the reviewer failed to see any novelty related to FPGA implementation.

6. The motivation for XNOR Signed Bits is unclear. The authors claim that it is for multiplying float with binary values (-1, 1). However, in IEEE float representation, the sign bit is implicit and it only needs to be changed from 0 to 1 or vice-versa. Detecting whether binary weight is -1 or 1 and simply flipping the 32nd or the 64th bit in float representation would change the sign. Why is the algorithm in Listing 1 needed? Also, Vivado HLS supports bit-accurate data type. Conversion between datatypes are not needed.

7. The section on experiments and results is limited to the authors' implementation in software and FPGA. This is a major concern with this paper as it makes no comparison with prior work. It is well known that FPGA implementations are typically targeted for hardware acceleration and hence they are supposed to be faster than CPU only implementation.

8. In Table 5, the latency of multiplication between half and int1 cannot be 0. There will be some delay associated with this operation. The absolute value of the delay can be less than 10 ns (100m MHz clock used in the design) and hence if registered inputs go to XSB and outputs of XSB operation are registered, the latency will be 1 clock cycle. Even if inputs and output are not registered, the absolute delay is there. Writing latency as 0 is not the correct way to interpret this delay.Half is 16 bits in representation. If one multiplies it with int1 i.e. 1 bit integer,the result is still 16 bits. It is not clear this resulted in only two FFs because a single FF can only store 1 bit.

9. Because of lack of comparison with prior/related work and no discussion of prior/related work, the list of references is incomplete.

10. Fig. 1 is incorrect. The XNOR result of 1 and 1 is 1 and not 0.

6. PLOS authors have the option to publish the peer review history of their article (what does this mean?). If published, this will include your full peer review and any attached files.

Reviewer #1: No

Reviewer #2: **Yes: **Tales C Pimenta

Reviewer #3: No

Reviewer #4: No

---

## [Author Response · Author response to Decision Letter 0]

3 Mar 2021

Dear reviewers,

We would like to inform that all number of page, reference, listing, and table in this letter referred to the Revised Manuscript with Track Changes.

Reply to reviewer 1:

Comment 1:

A comprehensive analysis, especially on larger and more complex networks such as the ResNet family, would be needed to provide a better understanding of the true value of the proposed scheme.

Reply 1: 

We conducted an additional experiment with the ResNet-18 model and CIFAR10 dataset in the CIFAR10 section (page 16-17). Using results from 70 rounds of Bayesian optimization (BO) search, we did a comprehension analysis on outcome combinations of mixed-precision weight network (MPWN). We found out that there are patterns of heuristic rules appeared in the top-5 combinations as shown in Table 6 (page 17).

Comment 2: 

For example, the authors devote a long discussion on “XNOR signed-bit”, which involves multiplying a floating-point activation with a binary weight. At the heart of this multiplication, essentially it is a simple implementation of changing the sign of a value by flipping the MSB bit of a floating-point number depending on the multiplied binary weight. A smarter synthesis tools could have easily performed that operation when instructed to multiply by -1. Therefore, the significance of the long discussion on how bypassing current limitation of one specific high-level synthesis tools is very limited.

Reply 2:

 We agree that by multiplying an input feature with a constant weight -1, the synthesis tool should only toggle only the signed-bit of the input feature. However, since this binary weight is a variable that can be in {0, -1}, this task becomes more complicated. With two possible states of int1 (1-bit integer), the smarter compiler may construct the XOR-signed bit (XSB) if it understands both binary representations of 16-bit floating point (half) and int1. The complier must also detect that only the signed-bit is changed in this multiplication (which may be weird from the multiplication with 0). Maybe there is the smart compiler that achieves both of these considerations, however, this task is not as simple as multiply by -1.

Comment 3: 

I suggest the authors to focus either on the optimization algorithms with more comprehensive network structures and datasets, or the authors may focus on the hardware implementations of different combination of network layers and show how they would have made the design more efficient than a fixed-precision network.

Reply 3:

Thank you for your suggestions, we revised with both your suggestions. We added the CIFAR10 section (page 16-17) that we discussed in Reply 1, and we also have revised our FPGA implementation of FBTBF to FTTTF (a better combination to FTTTT in our revised ASB) instead. Even though we did not implement the T layer with the sparse matrix multiplication or convolutional operation, we implemented T layer with a ternary bitwise operation (TBO) [30] instead. We discussed the TBO thoroughly in the ternary bitwise operation section (page 10-11). Using the same concept as XSB, the TBO designed for the ternary weight {-1, 0, 1} using only XOR and AND gates. As shown in Table 7 (page 18), TBO consumes only 0 latency and 32 FFs while the 32-bit floating point multiplication consumes 1 latency, 3 DSP48E, 130 FFs, and 150 LUTs.

Reply to reviewer 2

Comment 1:

 The article presenta a nice work of using mixed-precision weights networs (MPWN) to implement a quantization neral network. The approach is well described, as well the software simulation and hardware implementation.

Reply 1: 

Thank you for your kind response.

Reply to reviewer 3

Comment 1: 

The authors need to compare their results with other works in the literature. There are many other compression approaches based on quantization applied to the Deep Neural Networks in the literature. Which does the advantages of the proposal presented in this manuscript?

Reply 1: 

We provided the literature of the mixed-precision model in the last paragraph on page 2. Our main advantages compared with previous methods are we thoroughly explored the ASB metric with all possible combinations of the LeNet-5 model. Using this insight, we able to correct a negative correlation between the test accuracy and ASB using the min-max normalization and weighted average with ASB.

In terms of hardware implementation, we included a comparison with a prior work, Rongshi et al [39]. in Table 10 (page 20), 11 (page 21), and 12 (page 21). Rongshi et al. implemented a 32-bit floating-point LeNet-5 model on Xilinx Zybo Z7 board with Vivado HLS 2018.2. Our model performed better in terms of latency by 11.62 times with a trade-off with more hardware resources.

Comment 2: The authors used a High-Level Synthesis (HLS) tool to implement the proposal on FPGA. However, these kinds of tools insert a high overhead resource in the generated hardware. The authors must explain more about that. Does the HLS decrease the results performance? Does it not better to implement without HLS?

Reply 2:

We have added the explanation related HLS and HDL in the second paragraph on page 3 as follows:

“This allows users to rapidly develop applications with an interface of a C-like language with fewer constraints from the HDL language. With this property, HLS is especially useful when applies with the deep learning algorithm that its state-of-the-art algorithm has been being rapidly changed. However, a drawback of HLS compared to optimized handcrafted HDL is that HLS-generated HDL code may cause a higher latency and hardware utilization [21]. For instance, Ordaz et al. [21] compares HDL and HLS implementations of cyclic redundancy check (CRC). Ordaz et al. found out that HDL implementation consumes less than LUT by 1.58 times and less amount of latency by 2.63 times.”

Reply to reviewer 4

Comment 1.

 There are a number of papers in computer architecture and FPGA conferences as well as in ML/DL conferences/journals that deal with mixed precision networks. However, the current paper has no discussion of such prior work. Hence, it does not set the background for the current work properly. There is no comparison, qualitative and quantitative with such prior works either which is important to see where lies novelty in the current paper.

Reply 1:

We provided literatures of the mixed-precision model in the last paragraph on page 2. In this paragraph, to the best of our knowledge, we discussed our novelties comparing with the previous works as follows:

1. We utilized Bayesian optimization to search a suitable weight space instead of using the reinforcement learning or differentiable architecture search [20].

2. We also provide ASB scores that we specifically designed for the MPWN model.

3. We included a sparsity as a part of ASB and we left a choice to not quantization into the search space.

Comment 2.

 The motivation for using binary, ternary and FP weights in a neural network is not clear. It seems that the authors have based their decision based on their prior work [19]. However, the heuristics based prior work [19] does not explain how "large" or "small" is to be determined in the context of the current paper.

Reply 2:

 We have added definitions of "large" and "small" in the last paragraph of page 7 as follows:

“In this case, we define a large number as a number that more than one positive standard deviation from a mean and we define a small number as a number that less than or equal to one positive standard deviation from the mean.

Comment 3:

ASB is defined to be = (a+s+bnor)/3. However, the three quantities are not dimensionally equivalent. For instance, how is sparsity defined as a number - that should be explained and how dimensional analysis considerations are taken into account in this formula should be explained.

Reply 3: 

We have added a comprehension analysis on ASB from the second to the last paragraph of page 14. We found out that are differences in distributions between a, s, and b_nor. Each one of them does not contain a same range or variance. a contains especially low variance comparing with other two metrics due to a small range between minimum and maximum values of a. This causes the negative Pearson correlation between a and ASB. Due to this problem, we did some preprocessing and modifications to ASB as follows:

1. We rescaled each metric of ASB to a same range of [0, 1] by using a min-max normalization.

2. Using the min-max normalization requires actual minimum and maximum values of the distribution. However, this is not feasible without training all possible combinations of MPWN. Therefore, we proposed estimating rules to find estimated maximum and minimum values from the full precision, ternary and binary models instead.

3. To further balance the correlation between each parameter of ASB, we proposed to use ASB with a weighted average with weights: alpha, beta, and gamma.

By using a min-max normalization, we able to improve the Pearson correlation between a and ASB from the negative correlation -0.2249 to the positive correlation 0.2816.

Comment 4:

 FPGA implementation described in this paper is regular textbook implementation. Unfortunately, the reviewer failed to see any novelty related to FPGA implementation.

Reply 4: 

Our hardware novelty is mainly contributed from the XOR-signed bit (XSB). Using the same concept of XSB, we succeeded to implement a ternary bit-wise operation (TBO) [30] which we discussed in the Ternary bitwise operation section on page 10 and 11. Other than this is proof of work that our MPWN model can be implemented using both XSB and TBO.

Comment 5:

 The motivation for XNOR Signed Bits is unclear. The authors claim that it is for multiplying float with binary values (-1, 1). However, in IEEE float representation, the sign bit is implicit and it only needs to be changed from 0 to 1 or vice-versa. Detecting whether binary weight is -1 or 1 and simply flipping the 32nd or the 64th bit in float representation would change the sign. Why is the algorithm in Listing 1 needed? Also, Vivado HLS supports bit-accurate data type. Conversion between datatypes are not needed.

Reply 5: 

We have added a discussion on in the first paragraph of page 18. In terms of behavior, this should operate in a same manner as XSB. However, by "Detecting", this requires a control logic which is expensive in Vivado HLS. We did a comparison between XSB and Listing 3 (page 18) that utilizes the control logic by using C synthesis with Vivado HLS 2019.2. We found out that the Listing 3 consumes 2 cycles at 100 MHz, 2 DSPEs, 99 FFs, and 150 LUTs, while XSB consumes only 0 cycles (or in real-world 1 cycle) and 2 FFs.

Comment 6:

 The section on experiments and results is limited to the authors' implementation in software and FPGA. This is a major concern with this paper as it makes no comparison with prior work. It is well known that FPGA implementations are typically targeted for hardware acceleration and hence they are supposed to be faster than CPU only implementation.

Reply 6: 

In terms of the software, we did compare with both BinaryConnect (BC) [9] and Ternary weight networks (TWN) [11] without a scaling factor in Table 3 (page 14). In terms of hardware, we have added a comparison with prior work, Rongshi et al [39]. in the Table 10 (page 20), 11 (page 21), and 12 (page 21).

Comment 7:

. In Table 5, the latency of multiplication between half and int1 cannot be 0. There will be some delay associated with this operation. The absolute value of the delay can be less than 10 ns (100m MHz clock used in the design) and hence if registered inputs go to XSB and outputs of XSB operation are registered, the latency will be 1 clock cycle. Even if inputs and output are not registered, the absolute delay is there. Writing latency as 0 is not the correct way to interpret this delay.Half is 16 bits in representation. If one multiplies it with int1 i.e. 1 bit integer,the result is still 16 bits. It is not clear this resulted in only two FFs because a single FF can only store 1 bit.

Reply 7: 

 We only provided the latency that was reported from C synthesis with Vivado HLS. However, as you mentioned, the real latency cannot be 0, therefore we have added the following statement in the first paragraph of page 18.

“Note that in both XSB and TBO, the latency cannot be zero in practice. We only displayed the result from VHLS C synthesis.”

In the same paragraph, to address the FF problem, we have modified our analysis to the hardware resources of XSB as follows:

“XSB consumed only two FFs to perform multiplication, while TBO consumes 32 FFs. Each pair of FFs used to construct a logic gate. Therefore, XSB utilizes a logic gate, while TBO utilizes 16 logic gates.”

Comment 8: 

 Because of lack of comparison with prior/related work and no discussion of prior/related work, the list of references is incomplete.

Reply 8: 

We have added the discussion of prior works in the last paragraph of page 2. We have included the comparison with prior work, Rongshi et al [39]. in the Table 10 (page 20), 11 (page 21), and 12 (page 21).

Comment 9: 

Fig. 1 is incorrect. The XNOR result of 1 and 1 is 1 and not 0.

Reply 9: 

Thank you for mentioning this mistake. We corrected all "XNOR" to "XOR" throughout the paper.

---

## [Decision Letter · Decision Letter 1]

19 Mar 2021

PONE-D-20-36981R1

Mixed-precision weights network for field-programmable gate array

PLOS ONE

Dear Dr. Fuengfusin,

Thank you for submitting your manuscript to PLOS ONE. After careful consideration, we feel that it has merit but does not fully meet PLOS ONE’s publication criteria as it currently stands. Therefore, we invite you to submit a revised version of the manuscript that addresses the points raised during the review process.

We look forward to receiving your revised manuscript.

Kind regards,

Chi-Hua Chen, Ph.D.

Academic Editor

PLOS ONE

Reviewers' comments:

Reviewer's Responses to Questions

**Comments to the Author**

1. If the authors have adequately addressed your comments raised in a previous round of review and you feel that this manuscript is now acceptable for publication, you may indicate that here to bypass the “Comments to the Author” section, enter your conflict of interest statement in the “Confidential to Editor” section, and submit your "Accept" recommendation.

Reviewer #1: (No Response)

Reviewer #2: All comments have been addressed

Reviewer #4: (No Response)

2. Is the manuscript technically sound, and do the data support the conclusions?

Reviewer #1: Partly

Reviewer #2: Yes

Reviewer #4: Yes

3. Has the statistical analysis been performed appropriately and rigorously? 

Reviewer #1: N/A

Reviewer #2: Yes

Reviewer #4: N/A

4. Have the authors made all data underlying the findings in their manuscript fully available?

Reviewer #1: Yes

Reviewer #2: Yes

Reviewer #4: Yes

5. Is the manuscript presented in an intelligible fashion and written in standard English?

Reviewer #1: Yes

Reviewer #2: Yes

Reviewer #4: Yes

6. Review Comments to the Author

Reviewer #1: The authors have improved the manuscript from the previous submission and have addressed many of my earlier concerns. However, I remain skeptical about the novelty of the FPGA implementation, particularly the treatment regarding HLS. As some other reviewers have pointed out, the presented FPGA implementation of XSB and TBO are standard textbook techniques implementation of low bit width numerical operations. There is nothing wrong about these implementations, but at the same time I also do not see the novelty in their designs. As a result, the long discussion surrounding Listing 1 and Listing 2 in pages 9-10 seems fairly excessive to me and undermines the value of this paper. I believe the authors may simply lay out the hardware design as part of the implementation details.

To justify the novelty of the FPGA implementation, the authors should compare their implementations with other low-bitwidth (binary and ternary) DNN inference work, which is plentiful. The work of FINN, for example, allows binary inference network to be generated easily for FPGAs. There are many other similar frameworks for binary and ternary inference work on FPGA. The authors should compare their hardware implementation against them if they intend to claim novelty on their FPGA implementation. Otherwise, I believe the authors may simply focus on their novelty in deriving and utilising mixed precision implementations for inference. I think if the authors can demonstrate the benefit of mixed-precision network over standard fixed, but possibly low precision network, and if the authors can demonstrate how FPGAs are good target platform for such mixed-precision network as opposed to CPUs or GPUs solution, then the value of the manuscript would be much higher.

Reviewer #2: Thank you for the review. The paper is much better.

As I could observe, the comments of the reviewers ahve been met.

Reviewer #4: 1. The authors have tried to address most of my comments. However, I am not convinced with the response to my comment number 1 (Reviewer 4). Points 1, 2 and 3 listed in the author response to this comment are contributions. But are they novel when compared to prior work, especially in computer architecture and FPGA journal or conferences? The proof of novelty is not there. For instance, why is ASB so important a consideration for the authors?

2. The response to my comment 6 says that hardware results are compared with Rongshi et al [39]. But is that work the state-of-the-art work to make comparison with since comparison is being done only with one work?

3. The response to my comment 7 says "Each pair of FFs used to construct a logic gate". That is a fundamentally incorrect statement in digital logic and design. FFs cannnot be used to construct a logic gate. In fact, logic gates can be used to construct flip flops (FFs).

7. PLOS authors have the option to publish the peer review history of their article (what does this mean?). If published, this will include your full peer review and any attached files.

Reviewer #1: No

Reviewer #2: No

Reviewer #4: No

---

## [Author Response · Author response to Decision Letter 1]

6 Apr 2021

Dear reviewers,

We would like to inform that all number of page, reference, listing, and table in this letter referred to the Revised Manuscript with Track Changes.

Reply to reviewer 1:

Comment 1:

The authors have improved the manuscript from the previous submission and have addressed many of my earlier concerns. However, I remain skeptical about the novelty of the FPGA implementation, particularly the treatment regarding HLS. As some other reviewers have pointed out, the presented FPGA implementation of XSB and TBO are standard textbook techniques implementation of low bit width numerical operations. There is nothing wrong about these implementations, but at the same time I also do not see the novelty in their designs. As a result, the long discussion surrounding Listing 1 and Listing 2 in pages 9-10 seems fairly excessive to me and undermines the value of this paper.

Reply 1:

Thank you for your comments. We would like to address that even if our digital circuits, which are results from Listing 1 and 2, may not be novel designed circuits. One of our novelties in these listings realizes from how to bypass datatype constraints in Vivado High-level synthesis (VHLS). By converting half (half-precision floating-point) to ap_int (arbitrary-bit integer), we can access bitwise methods, and after we utilized all preferred bitwise operations, we can convert it back to half without any loss in information. This allows these digital circuits to be implement-able with VHLS and open a way to implements these circuits with a quantization neural network (QNN) with the floating-point activation and binary or ternary weights.

Comment 2:

I believe the authors may simply lay out the hardware design as part of the implementation details. To justify the novelty of the FPGA implementation, the authors should compare their implementations with other low-bitwidth (binary and ternary) DNN inference work, which is plentiful. The work of FINN, for example, allows binary inference network to be generated easily for FPGAs. There are many other similar frameworks for binary and ternary inference work on FPGA. The authors should compare their hardware implementation against them if they intend to claim novelty on their FPGA implementation. Otherwise, I believe the authors may simply focus on their novelty in deriving and utilising mixed precision implementations for inference. I think if the authors can demonstrate the benefit of mixed-precision network over standard fixed, but possibly low precision network.

Reply 2:

 We have added a section to indicate our novelties comparing with FINN [24] and other QNNs that were implemented to FPGA from Page 3 in the last paragraph. In general, to the best of our knowledge, our FPGA implementation of MPWN is the first FPGA implementation that utilizes binary or ternary weights with floating-point activations. We also displayed that XSB and TBO can be used to replace the multiplication processes in binary and ternary layers, respectively. Even though we did not provide a comparison with FINN, we provided comparisons with another binarized neural network (BNN) [10] framework, GUINNESS [25], which we are more familiar as shown in Tables 10, 11, and 12.

 We also included a comparison with BNN that contains both 1-bit weights and activations in Table 3. BNN promises a better alternative compared with MPWN in terms of hardware utilization. However, it performs worse in terms of test accuracy comparing with our models. 

Comment #3:

 If the authors can demonstrate how FPGAs are good target platform for such mixed-precision network as opposed to CPUs or GPUs solution, then the value of the manuscript would be much higher.

Reply #3:

 To address this point, we did a comparison between our FPGA implementation (Proposed directive) and a CPU ARM Cortex-53 on Table 8 on Page 19. Using TBO and FPGA parallelism, our FPGA implementation performs less latency than ARM Cortex-53 from 2.0 times to 11.77 times depend on weight layers.

Reply to reviewer 2

Comment #1:

 Thank you for the review. The paper is much better.

As I could observe, the comments of the reviewers ahve been met.

Reply #1:

 Thank you for your reviews. Your comments have improved this work.

Reply to reviewer 4

Comment #1 

 The authors have tried to address most of my comments. However, I am not convinced with the response to my comment number 1 (Reviewer 4). Points 1, 2 and 3 listed in the author response to this comment are contributions. But are they novel when compared to prior work, especially in computer architecture and FPGA journal or conferences? The proof of novelty is not there. For instance, why is ASB so important a consideration for the authors?

Reply #1:

Thank you for your comments. ASB is important to us because it is a metric that indicates the quality of our model and its ability to deploy into hardware devices. By defining this metric, we can utilize Bayesian optimization to automatically discover suitable combinations of weight layers to deploy into the FPGA.

We have added a section to indicate our novelties comparing with prior works in the FPGA field from Page 3 in the last paragraph. In general, to the best of our knowledge, our novelties in the FPGA field are we provide a first FPGA implementation of binary or ternary weights model with floating-point activation. To effectively deploy with binary or ternary weights and floating-point activations, we introduce XSB and TBO to replace floating-point multiplications with bitwise operations instead.

Comment #2

 The response to my comment 6 says that hardware results are compared with Rongshi et al [39]. But is that work the state-of-the-art work to make comparison with since comparison is being done only with one work?

Reply #2:

 The main reason that we have compared our work with Rongshi et al [26] is Rongshi et al. provides the same architecture (LeNet-5) with our work to compare with. For the same reason, we have added Cho et al [27] and GUINNESS [25] to compare with as shown in Tables 10, 11, and 12. Our model performs with less latency comparing than the prior works with a cost that our work consumes more hardware utilizations.

Comment #3

 The response to my comment 7 says "Each pair of FFs used to construct a logic gate". That is a fundamentally incorrect statement in digital logic and design. FFs cannnot be used to construct a logic gate. In fact, logic gates can be used to construct flip flops (FFs).

Reply #3:

 Thank you for mention this issue. We have checked the results and found out that we did typos from LUT to FF. We have adjusted our statement from “Each pair of FFs used to construct a logic gate.” to "Each pair of LUTs used to construct a logic gate." instead. We have corrected these mistakes throughout the paper.

---

## [Decision Letter · Decision Letter 2]

26 Apr 2021

Mixed-precision weights network for field-programmable gate array

PONE-D-20-36981R2

Dear Dr. Fuengfusin,

We’re pleased to inform you that your manuscript has been judged scientifically suitable for publication and will be formally accepted for publication once it meets all outstanding technical requirements.

Kind regards,

Chi-Hua Chen, Ph.D.

Academic Editor

PLOS ONE

Additional Editor Comments (optional):

Reviewers' comments:

Reviewer's Responses to Questions

**Comments to the Author**

1. If the authors have adequately addressed your comments raised in a previous round of review and you feel that this manuscript is now acceptable for publication, you may indicate that here to bypass the “Comments to the Author” section, enter your conflict of interest statement in the “Confidential to Editor” section, and submit your "Accept" recommendation.

Reviewer #1: All comments have been addressed

Reviewer #2: All comments have been addressed

Reviewer #4: All comments have been addressed

2. Is the manuscript technically sound, and do the data support the conclusions?

Reviewer #1: Yes

Reviewer #2: Yes

Reviewer #4: Yes

3. Has the statistical analysis been performed appropriately and rigorously? 

Reviewer #1: N/A

Reviewer #2: Yes

Reviewer #4: N/A

4. Have the authors made all data underlying the findings in their manuscript fully available?

Reviewer #1: (No Response)

Reviewer #2: Yes

Reviewer #4: No

5. Is the manuscript presented in an intelligible fashion and written in standard English?

Reviewer #1: (No Response)

Reviewer #2: Yes

Reviewer #4: Yes

6. Review Comments to the Author

Reviewer #1: The authors have addressed most of my previous comments. I do have my reservation on the novelty of the work, particularly regarding the FPGA implementations. However, I believe the manuscript is in an acceptable state.

Reviewer #2: The authors have added table 8, a comparison section, few coments and comparisons along the text.

My concerns have been met.

Reviewer #4: Comments have been addressed, more or less. The reviewer encourages authors to submit this kind of work to computer architecture or FPGA conferences in the future.

7. PLOS authors have the option to publish the peer review history of their article (what does this mean?). If published, this will include your full peer review and any attached files.

Reviewer #1: No

Reviewer #2: No

Reviewer #4: No

---

## [Editor Report · Acceptance letter]

29 Apr 2021

PONE-D-20-36981R2 

Mixed-precision weights network for field-programmable gate array 

Dear Dr. Fuengfusin:

I'm pleased to inform you that your manuscript has been deemed suitable for publication in PLOS ONE. Congratulations! Your manuscript is now with our production department. 

Kind regards, 

on behalf of

Professor Chi-Hua Chen 

Academic Editor

PLOS ONE